# TCAF1 promotes TRPV2-mediated Ca$^{2+}$ release in response to cytosolic DNA to protect stressed replication forks

Lingzhen Kong[1,5], Chen Cheng [1,5], Abigael Cheruiyot[1], Jiayi Yuan [1], Yichan Yang [1], Sydney Hwang [1], Daniel Foust [1], Ning Tsao[2], Emily Wilkerson[1], Nima Mosammaparast [2], Michael B. Major [1], David W. Piston [1], Shan Li [1,3,4] ✉ & Zhongsheng You [1] ✉

The protection of the replication fork structure under stress conditions is essential for genome maintenance and cancer prevention. A key signaling pathway for fork protection involves TRPV2-mediated Ca$^{2+}$ release from the ER, which is triggered after the generation of cytosolic DNA and the activation of cGAS/STING. This results in CaMKK2/AMPK activation and subsequent Exo1 phosphorylation, which prevent aberrant fork processing, thereby ensuring genome stability. However, it remains poorly understood how the TRPV2 channel is activated by the presence of cytosolic DNA. Here, through a genome-wide CRISPR-based screen, we identify TRPM8 channel-associated factor 1 (TCAF1) as a key factor promoting TRPV2-mediated Ca$^{2+}$ release under replication stress or other conditions that activate cGAS/STING. Mechanistically, TCAF1 assists Ca$^{2+}$ release by facilitating the dissociation of STING from TRPV2, thereby relieving TRPV2 repression. Consistent with this function, TCAF1 is required for fork protection, chromosomal stability, and cell survival after replication stress.

The cellular response to replication stress, which occurs frequently owing to the challenges posed by various intracellular and environmental factors, is crucial for the maintenance of genome stability and the prevention of human diseases such as cancer, ageing, and developmental disorders[1–3]. A key function of the replication stress response is the protection of replication fork structure from aberrant nucleolytic attack by DNases such as Mre11, Exo1, and Dna2, which otherwise would generate excessive ssDNA, resulting in DNA damage and chromosomal instability[4,5]. A number of factors such as BRCA1, BRCA2, FANCD2, BOD1L, RADX, and ZKSCAN3 have been shown to be important for fork protection, many of which act at the replication forks[6–12]. The ATR/Chk1-dependent checkpoint pathway also plays a critical role in fork protection, with Exo1 being a major target of suppression[13–18]. We have recently identified a separate Ca$^{2+}$-dependent signaling pathway that is also crucial for fork protection and cell survival under replication stress[19,20]. In this pathway, replication stress induces the generation of cytosolic DNA, which activates the sensor protein cGAS[21]. cGAS then catalyzes the synthesis of the second messenger cGAMP from GTP and ATP[22]. The binding of cGAMP to STING[23], which we find associates with and represses TRPV2 on the ER in the resting state, causes its dissociation from TRPV2, leading to TRPV2 derepression and Ca$^{2+}$ release. The resulting elevation of intracellular Ca$^{2+}$ (iCa$^{2+}$) then activates CaMKK2 and the downstream kinase AMPK[24–27]. Following activation, AMPK directly phosphorylates Exo1 at S746, leading to the binding of 14-3-3 proteins and sequestration of Exo1, thereby preventing abnormal processing of stalled replication

[1]Department of Cell Biology and Physiology, Washington University School of Medicine, St. Louis, MO 63110, USA. [2]Department of Pathology and Immunology, Washington University in St. Louis School of Medicine, St. Louis, MO 63110, USA. [3]Zhejiang Provincial Key Laboratory of Pancreatic Disease in the First Affiliated Hospital, Zhejiang University School of Medicine, Hangzhou, Zhejiang 310029, China. [4]Cancer Center, Zhejiang University, Hangzhou, Zhejiang 310029, China. [5]These authors contributed equally: Lingzhen Kong, Chen Cheng. ✉e-mail: shan.li@zju.edu.cn; zyou@wustl.edu

forks[19,20,28]. Disruption of this signaling pathway causes excessive ssDNA, chromosomal instability, and compromised cell viability in the presence of replication stress[19,20]. However, despite the central role of the TRPV2-mediated $Ca^{2+}$ release in the activation of the genome protection pathway, it remains unclear how STING dissociates from TRPV2 and how TRPV2 is activated for $iCa^{2+}$ elevation under replication stress or other conditions that cause cGAS/STING activation.

To address this fundamental question, we carried out a genome-wide CRISPR/Cas9 screen in an effort to identify additional factors that regulate TRPV2 in the $Ca^{2+}$-dependent pathway required for fork protection and cell survival in the presence of replication stress. This led to the identification of TCAF1 (also named FAM115A) as a potential fork protection factor and a potential regulator of TRPV2 in the signaling pathway. As a relatively understudied protein, TCAF1 was recently identified as a regulator of TRPM8, a member of the TRP channel family that is activated by low temperature and menthol[29]. TCAF1 binds to TRPM8 and positively modulates its channel activity. Additionally, TCAF1 promotes the trafficking of TRPM8 to the cell surface in response to menthol treatment. Furthermore, TCAF1 has been shown to interact with other TRP channels, such as TRPV6 and TRPM2[29]. These observations and our screen results suggest that TCAF1 may also regulate TRPV2 in the $Ca^{2+}$-dependent fork protection pathway in the replication stress response. Indeed, here we show that TCAF1 is a novel fork protection factor that promotes TRPV2-dependent $iCa^{2+}$ elevation after replication stress. Mechanistically, TCAF1 facilitates the dissociation between STING and TRPV2 in response to cytosolic DNA or other conditions that activate cGAS/STING, thereby promoting $Ca^{2+}$ release from the ER.

## Results

### TCAF1 promotes cell survival after replication stress
We previously showed that replication stress induces cytosolic DNA, which triggers the release of $Ca^{2+}$ from the ER through TRPV2 and the activation of the downstream pathway to promote replication fork protection and cell survival[19,20]. To further decipher the mechanism for the TRPV2-mediated $Ca^{2+}$ release in the pathway, we performed a genome-wide CRISPR/Cas9 knockout screen to identify factors that promote cell survival under replication stress by suppressing Exo1-mediated fork processing. We reasoned that potential ion channel regulators among the identified factors may regulate TRPV2 in the $Ca^{2+}$-dependent fork protection pathway. To do the screening, we generated a wild-type (WT) HeLa cell line and an Exo1-KO HeLa cell line stably expressing Cas9. Both cell lines were then infected with lentiviruses expressing the pooled GeCKOv2 sgRNA library, which was designed to have 6 sgRNAs each for the 19,050 protein-encoding genes in the human genome[30]. Ten days after infection, the cells were treated with the replication stressor hydroxyurea (HU) or $H_2O$ for 24 hours. Three days after treatment, surviving cells were collected, and genomic DNA was isolated from all four samples. The integrated sgRNA inserts in the genomic DNA were then amplified by PCR. After the addition of Illumina sequencing adapters also via PCR, sgRNA inserts were subjected to Next-Gen sequencing to obtain read counts for each sgRNA in the library. MAGeCK analysis was then performed to rank genes based on the extent of depletion of their respective sgRNAs in the HU-treated WT sample[31]. Among the high-ranked genes, the ion channel regulator TCAF1 caught our attention because two of the three sgRNAs that were detected in the sequenced library exhibited deep depletion in the HU-treated WT sample, which was partially rescued by Exo1-KO (Fig. 1A, Supplementary Data 1 and 2). The third sgRNA detected in the sequenced library showed little depletion in WT cells after HU treatment, and no rescue (instead a small exacerbation) effect of Exo1-KO was observed for this sgRNA (Fig. 1A), likely because the sgRNA was much less efficient in deleting TCAF1 in cells. The screen result raises the possibility that TCAF1 promotes cell survival after replication stress by suppressing Exo1 function. We have previously

shown that TRPV2-mediated $Ca^{2+}$ signaling is critical for restricting the aberrant EXO1 activity for fork protection under replication stress. Intriguingly, TCAF1 was previously shown to interact with multiple TRP channels, including TRPM8, TRPV6, and TRPM2, and positively regulate the activity of TRPM8[29]. These observations and our screen result motivated us to investigate the potential role of TCAF1 in regulating TRPV2 in the $Ca^{2+}$-dependent fork protection pathway.

To determine whether TCAF1 regulates TRPV2 in fork protection, we first set out to validate the screen result by performing a clonogenic assay to determine the requirement of TCAF1 for cell survival after replication stress. We used a 3' UTR-targeting shRNA in a lentiviral vector to silence TCAF1 in HeLa cells and examined its effects on cell survival after HU treatment. To demonstrate the specificity of the knockdown effects, we ectopically expressed Flag-tagged TCAF1 (Flag-TCAF1) as a means to rescue the phenotypes of TCAF1-depleted cells. To detect endogenous TCAF1 protein in cells, we generated polyclonal antibodies against human TCAF1 in rabbits. As shown in Fig. 1B, both endogenous and exogenously expressed TCAF1 exhibit two specific bands on the western blot. These two bands were also detected in N-terminally 3xFlag-tagged and C-terminally HA-tagged TCAF1 (3xFlag-TCAF1-HA) by antibodies against either tag, suggesting this double-band pattern was not a result of protein cleavage in cells (Supplementary Fig. 1A). Using these tools, we found that TCAF1 knockdown caused HU sensitivity in cells, confirming the CRISPR/Cas9 screen result (Fig. 1B). Importantly, this HU-sensitivity was rescued by Flag-TCAF1 expression (Fig. 1B). These results suggest that TCAF1 possesses a previously unrecognized function in promoting cell survival after replication stress. In contrast to the HU treatment, TCAF1-depletion did not obviously affect the cell sensitivity to the radio-mimetic drug bleomycin and the DNA topoisomerase I inhibitor camptothecin (CPT) (Supplementary Fig. 1B), suggesting TCAF1 plays a distinct role in cell survival after replication stress.

### TCAF1 is required for replication fork protection and chromosome stability after replication stress
We hypothesized that TCAF1 regulates TRPV2 in fork protection and cell survival after replication stress. To test this idea, we first measured replication fork resection following TCAF1 silencing, TCAF1 CRISPR-based KO, and TCAF1 rescue. We used a native BrdU immuno-fluorescence (IF) assay to detect ssDNA levels in cells and a DNA fiber assay to measure nascent DNA degradation. Both assays are well-established and commonly used for detecting replication fork resection[19,20,32,33]. Using these methods, we found that shRNA-mediated TCAF1 knockdown indeed caused a higher level of fork resection in HeLa cells after HU treatment (Figs. 1C and D). A similar result was obtained in U2OS cells (Supplementary Fig. 1C). A TCAF1-KO HeLa cell line generated via CRISPR/Cas9 also exhibited a higher level of fork resection after HU treatment compared to the parental cell line (Supplementary Fig. 1D). This phenotype, again, was rescued by ectopic expression of Flag-TCAF1 (Supplementary Fig. 1D). No obvious changes in the cell cycle were observed in TCAF1-depleted cells (Supplementary Fig. 1E), indicating that the fork resection phenotype of TCAF1 described above was not an indirect effect of cell cycle alterations. Consistent with its function in fork protection, TCAF1 depletion also caused a marked increase in chromosomal instability in HU-treated cells, as measured by a chromosome spreading assay (Fig. 1E). Taken together, these data suggest that TCAF1 plays an important role in fork protection, chromatin stability maintenance, and cell survival in the presence of replication stress.

### TCAF1 prevents Exo1-mediated excessive fork resection by promoting the activation of the TRPV2-$Ca^{2+}$-CaMKK2-AMPK pathway
To define the molecular function of TCAF1 in fork protection, we tested whether the fork resection caused by TCAF1 depletion resulted

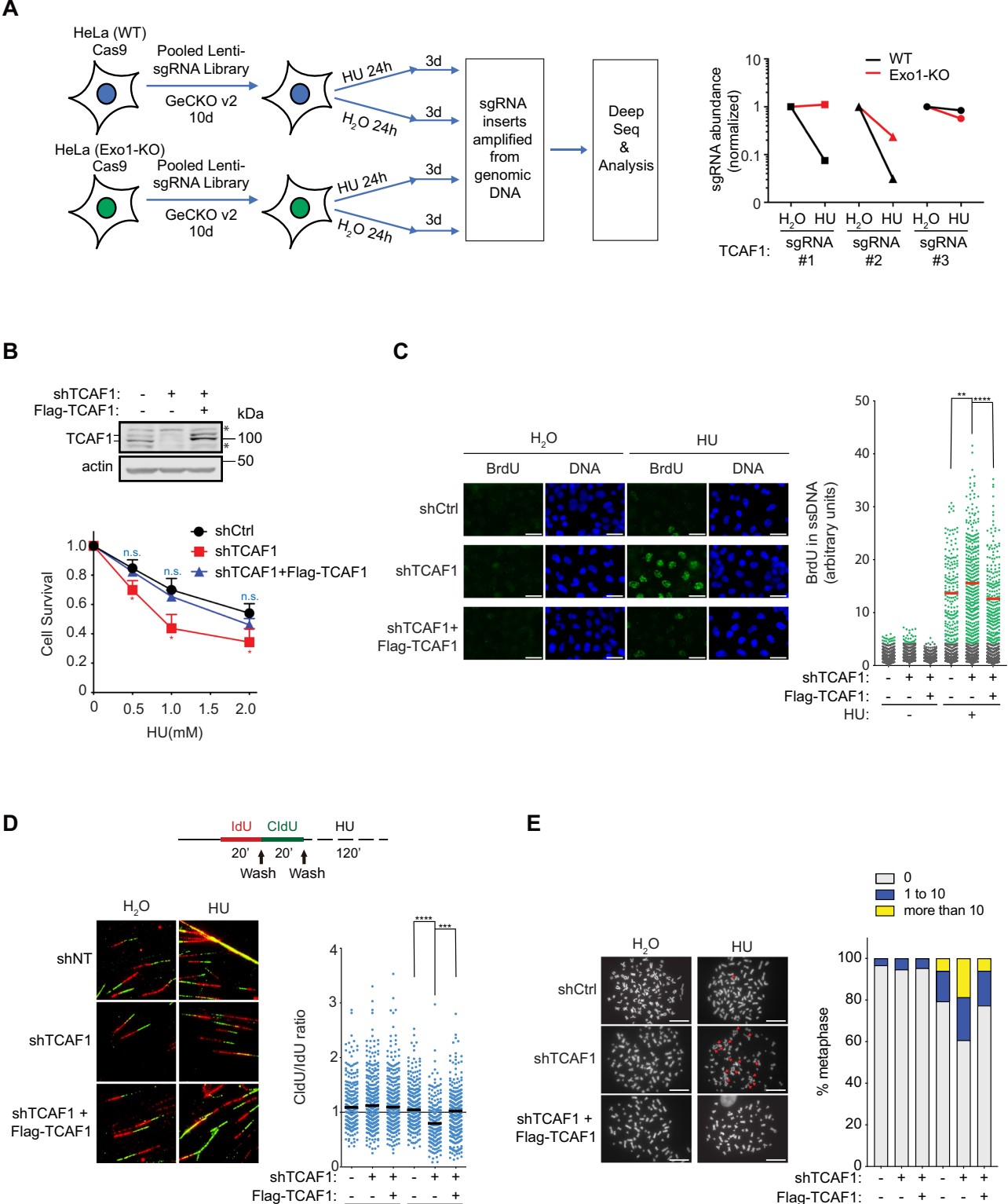

from uncontrolled Exo1 activity. Indeed, siRNA-mediated knockdown of Exo1 in TCAF1-depleted cells completely rescued the fork resection phenotype in HU-treated cells, suggesting that TCAF1 functions to suppress Exo1 to prevent fork resection after replication stress (Fig. 2A). In further support of this idea, Exo1 knockdown also largely rescued the chromosomal instability and HU-sensitivity phenotypes of TCAF1-depleted cells after replication stress (Figs. 2B and C). We also examined whether Mre11 and DNA2, two other fork resection

nucleases, are involved in the fork resection phenotype in TCAF1-depleted cells[9,34–36]. As shown in Supplementary Fig. 2A, Mirin-mediated inhibition of Mre11 also reversed the fork resection phenotype of TCAF1-depleted cells after HU treatment. This result is consistent with the notion that Mre11 mediates the initial cleavage of stalled replication forks prior to long-range resection by Exo1[37]. In contrast to Mre11 and Exo1, siRNA-mediated knocking down of DNA2 did not prevent aberrant fork resection in TCAF1-depleted cells in the

**Fig. 1 | TCAF1 is required for replication fork protection, chromosome stability maintenance, and cell survival upon replication stress. A** Left panel: Experimental scheme of a genome-wide CRISPR knockout screen for identifying new factors that promote cell survival in the presence of replication stress by suppressing Exo1 function. Right panel: Changes in the relative abundance of the three TCAF1 sgRNAs detected in the sgRNA library after HU treatment (4 mM, 24 h) in WT and Exo1-KO HeLa cells. The relative abundance of the sgRNAs in $H_2O$-treated samples was normalized to 1. Note that the other three sgRNAs for TCAF1 in the original library were not detected by deep sequencing. **B** Upper panel: Western blot analysis of TCAF1 knockdown by a UTR-targeting shRNA (#2) and the expression of Flag-TCAF1 in TCAF1-knockdown HeLa cells. *, nonspecific bands. Bottom panel: Effects of TCAF1 knockdown on cell survival and its rescue by Flag-TCAF1 expression after treatment with indicated concentrations of HU for 24 h. Data represent mean ± S.D. from triplicates. *, $p \le 0.05$ (two-tailed, unpaired t-test). See the source data for the exact $P$ values. **C** Effects of TCAF1 knockdown on replication fork resection detected using a native BrdU IF assay, and the rescue of fork resection by Flag-TCAF1 expression after HU treatment (2 mM, 5 h). Left panel: Representative BrdU IF images for the indicated samples (scale bar, 25 μm). Right panel: Quantified BrdU signal. Cells with a BrdU signal higher than the majority (98%) of $H_2O$-treated

control cells (gray dots) were taken as BrdU-positive (green dots). Red bars represent the mean BrdU intensity of BrdU-positive cells. At least 1,000 cells were analyzed for each sample. $n = 3$, ****$p \le 0.0001$. **$p \le 0.01$(two-tailed, unpaired t-test). See the source data for the exact $P$ values. Outlier signals were removed through ROUT (Q = 1%) analysis. **D** Effects of TCAF1 knockdown on fork resection detected using a DNA fiber assay, and the rescue of fork resection by Flag-TCAF1 expression after HU treatment (4 mM, 2 h). Upper panel: Experimental scheme (see Methods). Bottom left panel: representative images of DNA fibers for the indicated samples. Bottom right panel: dot plot of the CIdU/IdU track lengths ratio. Black bars represent the median. At least 200 tracks were scored for each sample. ****$p \le 0.0001$. ***$p \le 0.001$ (two-tailed, unpaired t-test). See the source data for the exact $P$ values. **E** Effects of TCAF1 knockdown on chromosomal integrity analyzed using a metaphase spreading assay, and the rescue of chromosomal integrity by Flag-TCAF1 expression after HU treatment (4 mM, 4 h). Left panel: Representative images of metaphase chromosome spreads for the indicated samples (scale bar, 25 μm). Chromosomal aberrations are marked by arrows. Right panel: Quantified result of the samples depicted in the left panel. 150 metaphases from three independent experiments were examined for each sample. Source data are provided as a Source Data file.

---

presence of HU (Supplementary Fig. 2B), consistent with the idea that DNA2 and Mre11/Exo1 are regulated by different factors in fork resection[34,38]. Together, these data strongly suggest that TCAF1 protects against Mre11/Exo1-mediated fork processing under replication stress.

We next determined whether TCAF1 acts in the TRPV2-$Ca^{2+}$-CaMKK2-AMPK signaling pathway, which is known to protect stressed replication forks from Exo1-mediated aberrant resection[19,20]. To this end, we first examined whether TCAF1 is required for TRPV2-mediated $Ca^{2+}$ release after replication stress. Using two biosensors GCaMP6s and GCaMPer, which measure intracellular and intra-ER $Ca^{2+}$ levels, respectively[39,40], we previously showed that HU treatment causes a TRPV2-mediated $Ca^{2+}$ release from the ER. This results in an elevation in intracellular $Ca^{2+}$ ($iCa^{2+}$) that is required for downstream CaMKK2/AMPK activation and fork protection[19,20,39,40]. Remarkably, we found that shRNA-mediated TCAF1 knockdown abrogated the ER $Ca^{2+}$ release and $iCa^{2+}$ elevation in HeLa cells expressing the $Ca^{2+}$ biosensors (Figs. 2D and E). Similar results were also obtained in U2OS cells and MCF10A cells (Supplementary Fig. 2C-E). These data strongly suggest that TCAF1 is required for TRPV2-mediated $Ca^{2+}$ release after replication stress. In further support of this idea, we found that knockdown of TCAF1 in HeLa cells expressing a dominant negative mutant of TRPV2 (TRPV2(DN))[41], which blocks $Ca^{2+}$ release, did not cause further reduction of $iCa^{2+}$ levels after HU treatment, suggesting that TCAF1 and TRPV2 function in the same pathway (Supplementary Fig. 2F). Consistent with its role in regulating $Ca^{2+}$ release, TCAF1 knockdown also abolished T172-phosphorylation of AMPK by CaMKK2 in multiple cell lines, a functional marker for AMPK activation (Fig. 2F, Supplementary 2G and 2H)[19]. TCAF1 depletion, however, did not affect ATR-mediated S345-phosphorylation of Chk1 after HU treatment, suggesting that it is not required for the activation of the checkpoint pathway (Fig. 2F and Supplementary 2H). This is also in agreement with our previous finding that the ATR/Chk1 checkpoint and the $Ca^{2+}$/AMPK-dependent signaling pathways operate separately in the replication stress response. Taken together, these results indicate that TCAF1 functions in the TRPV2-$Ca^{2+}$-CaMMK2-AMPK signaling pathway to protect replication forks from aberrant processing by Exo1.

## TCAF1 promotes $Ca^{2+}$ release in response to cytosolic DNA or direct cGAS activation

We have previously shown that replication stress-induced ER release of $Ca^{2+}$ is triggered by cytosolic DNA through a series of signaling events, including cGAS activation, cGAMP synthesis, cGAMP binding to STING, and STING-TRPV2 dissociation[20]. Other conditions that cause cytosolic

DNA or direct cGAS activation can also activate the signaling pathway[20]. To further test the role of TCAF1 in the TRPV2/$Ca^{2+}$ pathway, we generated cytosolic DNA by depleting TREX1, a nuclease that degrades cytosolic DNA, or by directly transfecting plasmid DNA into cells, both of which induce TRPV2-mediated $Ca^{2+}$ release[20]. We found that shRNA-mediated knockdown of TCAF1 blocked ER $Ca^{2+}$ release and $iCa^{2+}$ elevation induced by TREX1 depletion or DNA transfection, as measured by the GCaMP6s and GCaMPer biosensors (Fig. 3A-D). These results further demonstrate the function of TCAF1 in the TRPV2/$Ca^{2+}$ pathway.

Because cytosolic DNA sensing by cGAS induces TRPV2-mediated $Ca^{2+}$ release from the ER, we next determined whether direct cGAS activation can also induce $Ca^{2+}$ release and whether TCAF1 is required for this release[20]. To this end, we treated cells with $MnCl_2$, which has been reported to directly bind and activate cGAS to induce an innate immune response in the absence of cytosolic DNA[42,43]. Using the GCaMPer reporter, we found that $MnCl_2$ treatment indeed induced $Ca^{2+}$ release from the ER (Fig. 3E). Importantly, this $Ca^{2+}$ release is also dependent on TRPV2, as overexpression of TRPV2(DN) fully blocked the release (Supplementary Fig. 3A). TCAF1 depletion also largely abrogated ER $Ca^{2+}$ release induced by $MnCl_2$ (Fig. 3E), indicating that TCAF1 functions downstream of cGAS in the pathway. $MnCl_2$ treatment also induced a marked elevation of $iCa^{2+}$ (Fig. 3F). However, this $iCa^{2+}$ elevation was only partially abolished by TCAF1 depletion or by TRPV2(DN) overexpression (Fig. 3F and Supplementary 3B), suggesting that other ion channels were affected by $MnCl_2$ treatment in addition to TRPV2. Consistent with its effects on ER $Ca^{2+}$ release and $iCa^{2+}$ elevation, $MnCl_2$ treatment also caused AMPK phosphorylation, which was partially abrogated by TCAF1 knockdown or TRPV2(DN) overexpression (Fig. 3G and Supplementary 3C). Importantly, we found that TCAF1 depletion and TRPV2(DN) overexpression did not cause an additive effect on $iCa^{2+}$ elevation after $MnCl_2$ treatment, further suggesting that TCAF1 and TRPV2 function in the same pathway in regulating $Ca^{2+}$ release (Fig. 3H). Taken together, these results support a model wherein TCAF1 promotes TRPV2-mediated $Ca^{2+}$ release after cGAS activation in response to cytosolic DNA.

## TCAF1 acts downstream of cGAMP to relieve STING-mediated TRPV2 repression

In the cytosolic DNA-elicited $Ca^{2+}$ signaling pathway, cGAS-synthesized cGAMP binds to STING on the ER, leading to its dissociation from TRPV2, which in turn causes TRPV2 derepression and $Ca^{2+}$ release[20]. To determine whether TCAF1 acts upstream or downstream of cGAMP in the pathway, we first determined whether

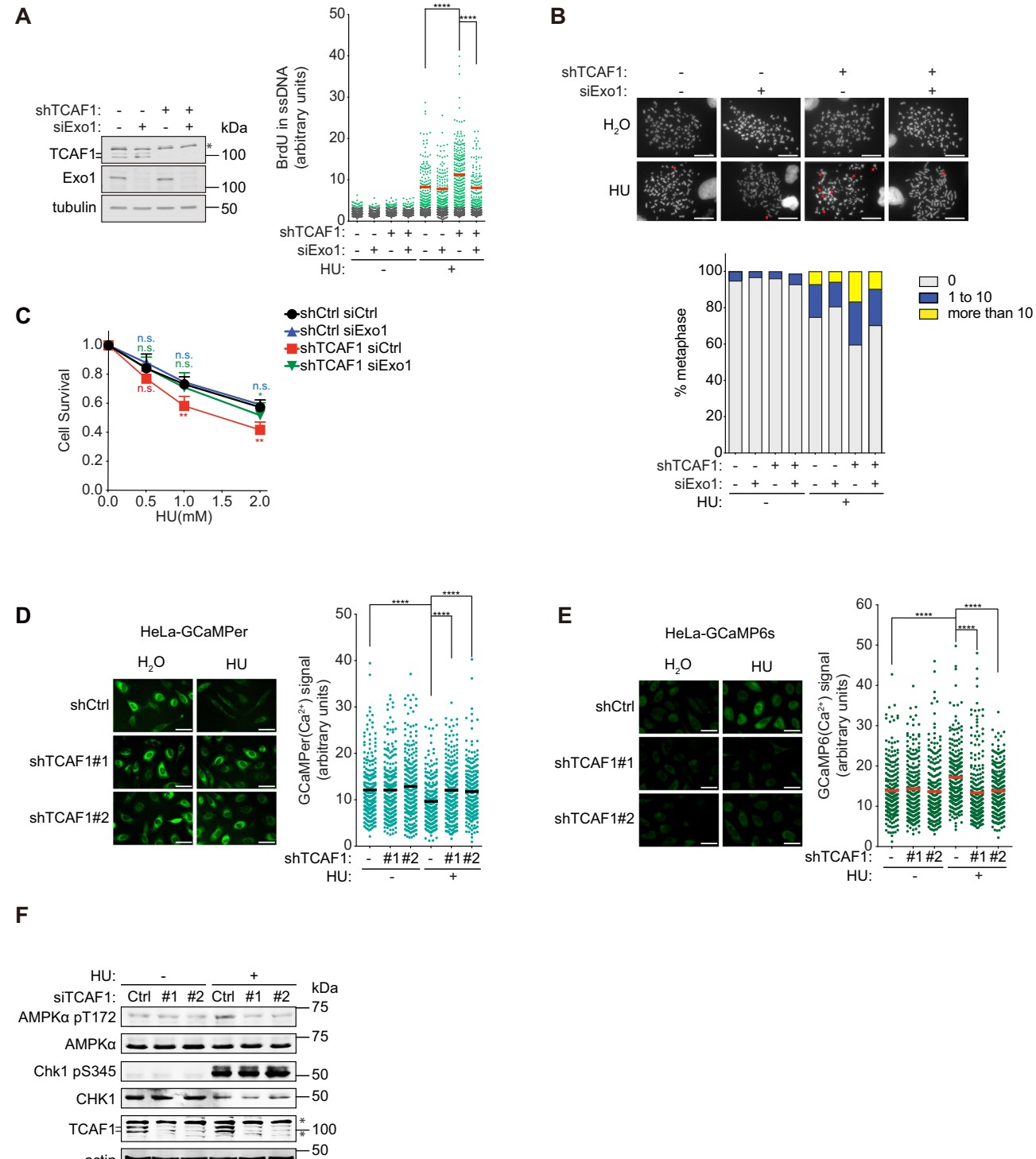

TCAF1 is required for cGAMP production after DNA transfection (to introduce cytosolic DNA). As a control, we depleted cGAS from cells, which is expected to abolish cGAMP production. Using an ELISA method to directly measure cGAMP levels in cells, we found that cGAS depletion abrogated cGAMP synthesis in DNA-transfected HeLa cells, as expected (Fig. 4A). In contrast, TCAF1 knockdown had little or no effect on cGAMP production in the presence of cytosolic DNA, suggesting that TCAF1 acts downstream of cGAMP in the $Ca^{2+}$-dependent signaling pathway (Fig. 4A). In further support of this idea, we found that TCAF1 depletion prevented ER $Ca^{2+}$ release and $iCa^{2+}$ elevation induced by direct cGAMP transfection (Figs. 4B and C).

We have previously shown that STING depletion is sufficient in inducing TRPV2-mediated $Ca^{2+}$ release in the absence of cytosolic DNA[20]. To determine whether TCAF1 plays a role in activating TRPV2 induced by STING depletion, we knocked down STING using a shRNA in control-knockdown or TCAF1-knockdown cells expressing GCaMP6s. Interestingly, TCAF1 knockdown did not affect this $iCa^{2+}$ elevation induced by STING depletion (Fig. 4D), suggesting that TCAF1 is no longer needed for $Ca^{2+}$ release when the STING-mediated TRPV2 repression is already relieved. Taken together, these data strongly suggest that TCAF1 functions downstream of cGAMP to promote the release of STING-mediated TRPV2 repression in the $Ca^{2+}$-dependent signaling pathway.

**Fig. 2 | TCAF1 protects replication forks under stress through the $Ca^{2+}$-CaMKK2-AMPK-Exo1 signaling pathway. A** Left panel: siRNA-mediated knockdown of Exo1 in control- and TCAF1-knockdown HeLa cells. Right panel: Effects of Exo1 knockdown on fork resection in control-knockdown and TCAF1-knockdown cells treated with HU (2 mM, 5 h) or $H_2O$. Red bars represent the mean BrdU intensity of BrdU-positive cells. At least 1,000 cells were analyzed for each sample. $n = 3$, ****, $p \leq 0.0001$ (two-tailed, unpaired t-test). See the source data for the exact $P$ values. **B** Effects of Exo1 knockdown on chromosomal stability in control-knockdown and TCAF1-knockdown HeLa cells after HU (4 mM, 4 h) or $H_2O$ treatment. Top panel: Representative images of metaphase chromosome spreads for the indicated samples (scale bar, 25 μm). Chromosomal aberrations are marked by arrows. Bottom panel: Quantified result of the samples depicted in the top panel. 150 metaphases from three independent experiments were examined for each sample. **C** Effects of Exo1 knockdown on the clonogenic survival of control-knockdown and TCAF1-knockdown HeLa cells treated with indicated concentrations of HU for 24 h. Data

represent mean ± S.D. from triplicates. **, $p \leq 0.01$ (two-tailed, unpaired t-test). See the source data for the exact $P$ values. **D** Effects of TCAF1 knockdown on HU-induced ER $Ca^{2+}$ release in HeLa cells. Left panel: Representative images of the GCaMPer signal (scale bar, 25 μm). Right panel: Quantified GCaMPer signal in S phase-synchronized cells treated with HU (4 mM, 4 h) or $H_2O$. 250 cells were scored for each sample. Black bars represent the mean. $n = 3$, ****, $p \leq 0.0001$ (two-tailed, unpaired t-test). See the source data for the exact $P$ values. **E** Effects of TCAF1 knockdown on HU-induced $iCa^{2+}$ elevation in HeLa cells. Left panel: Representative images of the GCaMP6s signal (scale bar, 25 μm). Right panel: Quantified GCaMP6s signal in S phase-synchronized cells treated with HU (4 mM, 4 h) or $H_2O$. 250 cells were scored for each sample. Red bars represent the mean. $n = 3$, ****, $p \leq 0.0001$ (two-tailed, unpaired t-test). See the source data for the exact $P$ values. **F** Effects of TCAF1 knockdown on HU-induced T172-phosphorylation of AMPKα and S345-phosphorylation of Chk1 in HeLa cells treated with HU (4 mM, 6 h) or $H_2O$. Source data are provided as a Source Data file.

## TCAF1 facilitates STING-TRPV2 dissociation in response to cytosolic DNA or direct cGAS activation

We have previously shown that upon cGAMP binding, STING dissociates from TRPV2, which is required for TRPV2 derepression and $Ca^{2+}$ release[20]. To further define the function of TCAF1 in the pathway, we determined whether it is required for the STING-TRPV2 dissociation. To this end, we first used a proximity ligation assay (PLA) to detect the association between STING-HA and TRPV2-Flag that were stably expressed in HeLa cells. As shown before, STING and TRPV2 exhibited a strong PLA signal in the absence of cytosolic DNA[20]. Direct DNA transfection dramatically reduced the PLA signal, indicative of the dissociation between the two proteins in the presence of cytosolic DNA (Fig. 5A)[20]. Remarkably, TCAF1 knockdown at least partially rescued the PLA signal in the presence of transfected DNA, suggesting that TCAF1 is important for the STING-TRPV2 dissociation (Fig. 5A). In further support of this idea, we found that direct cGAMP transfection or HU treatment also reduced the STING-TRPV2 PLA signal and that TCAF1 knockdown also partially rescued the PLA signal (Fig. 5B and Supplementary 4A). To further demonstrate the role of TCAF1 in STING-TRPV2 dissociation, we performed co-immunoprecipitation (co-IP) experiments for STING-HA and TRPV2-Flag in HeLa cells in the presence or absence of transfected DNA. As shown in Fig. 5C, DNA transfection markedly reduced the association between these two proteins, which again, was partially rescued by TCAF1 knockdown. Similarly, direct cGAS activation by $MnCl_2$ also caused the dissociation between STING-HA and TRPV2-Flag, which, again, was partially reversed by TCAF1 depletion (Supplementary Fig. 4B). These data strongly suggest that TCAF1 promotes the dissociation of STING from TRPV2 after cGAS activation and cGAMP production.

To begin to define the mechanism by which TCAF1 promotes STING-TRPV2 dissociation, we examined the potential interactions between TCAF1 and these two proteins. The results of co-IP experiments indicate that TCAF1 associates with both STING and TRPV2 in cells (Fig. 5D). Interestingly, we found that while STING knockdown did not affect the TCAF1-TRPV2 association, TRPV2 depletion partially abrogated the TCAF1-STING association (Figs. 5E and F). This suggests the possibility that the association between TCAF1 and STING is at least partially bridged by TRPV2. In further support of this idea, we found that STING(FTW/AAS), which is deficient in TRPV2 interaction (Figure S4C)[20], exhibited reduced association with TCAF1, compared with STING(WT) (Fig. 5G). To further elucidate the nature of TCAF1's interactions with TRPV2 and STING, we examined the potential changes of the interactions upon cytosolic DNA induction or direct cGAS activation. The results of PLA experiments indicate that the TCAF1-STING association was markedly reduced after plasmid DNA transfection or $MnCl_2$ treatment (Fig. 5H and Supplementary 4D). This is similar to TRPV2, which also dissociates from STING in response to cytosolic DNA or cGAS activation[20]. In contrast to the TCAF1-STING

association, little or no change was observed for the TCAF1-TRPV2 association after DNA transfection (Fig. 5I). Taken together, these data suggest that TCAF1 functions as a stable partner of TRPV2 and that upon cGAS activation it facilitates the dissociation between TRPV2 and STING to promote TRPV2-mediated $Ca^{2+}$ release.

## Discussion

This study has identified TCAF1 as a novel genome protection factor, which functions to facilitate TRPV2-mediated $Ca^{2+}$ release to promote replication fork protection and cell survival in the presence of replication stress. Our findings provide compelling evidence that TCAF1 protects replication forks from aberrant processing by Exo1 by promoting the activation of the cytosolic DNA/$Ca^{2+}$-dependent signaling pathway. Mechanistically, TCAF1 facilitates the TRPV2-mediated release of $Ca^{2+}$ from the ER by promoting the STING-TRPV2 dissociation, likely through its direct interactions with the two proteins (Fig. 6).

The identification of TCAF1 as a novel genome protection factor and TRPV2 regulator broadens our understanding of its molecular functions and sheds new light on its cancer relevance. Gkika et al. have previously shown that TCAF1 interacts with and regulates the activity of the $Ca^{2+}$ channel TRPM8 and its cell surface expression upon menthol treatment. In addition, the authors have found that TRPM8 and TCAF1 are upregulated in primary prostate cancer samples and downregulated in metastatic samples. This downregulation is apparently important for cell migration, a prerequisite for metastasis[29,44]. The results of our study suggest that TCAF1 may also modulate tumorigenesis by promoting genome maintenance and cell survival after replication stress, which occurs frequently due to oncogene activation or tumor suppressor gene inactivation[45,46]. By suppressing mutations and genomic instability in normal cells, TCAF1 may serve to suppress cancer initiation. However, by protecting the genome of cancer cells and promoting their survival in the presence of replication stress, TCAF1 may promote cancer progression. Thus, TCAF1 may be targeted for treating cancers with intrinsic or induced replication stress.

Our findings also provide new insights into the intricate mechanisms that regulate TRPV2 and $Ca^{2+}$ release triggered by cytosolic DNA and cGAS/STING activation. In the resting state, STING associates with and represses TRPV2 to prevent unscheduled $Ca^{2+}$ release. Upon replication stress or other conditions that induce cytosolic DNA, the binding of cGAMP to STING causes its dissociation from TRPV2, leading to TRPV2 derepression and $Ca^{2+}$ release from the ER[20]. We found in this study that TCAF1 is important for this dissociation and subsequent $Ca^{2+}$ release and pathway activation. Our results suggest that TCAF1 functions as a physical and functional partner of TRPV2 and that they dissociate from STING together upon cGAMP binding to STING, allowing $Ca^{2+}$ release from the ER. The function of TCAF1 in TRPV2 regulation is apparently limited to counteracting

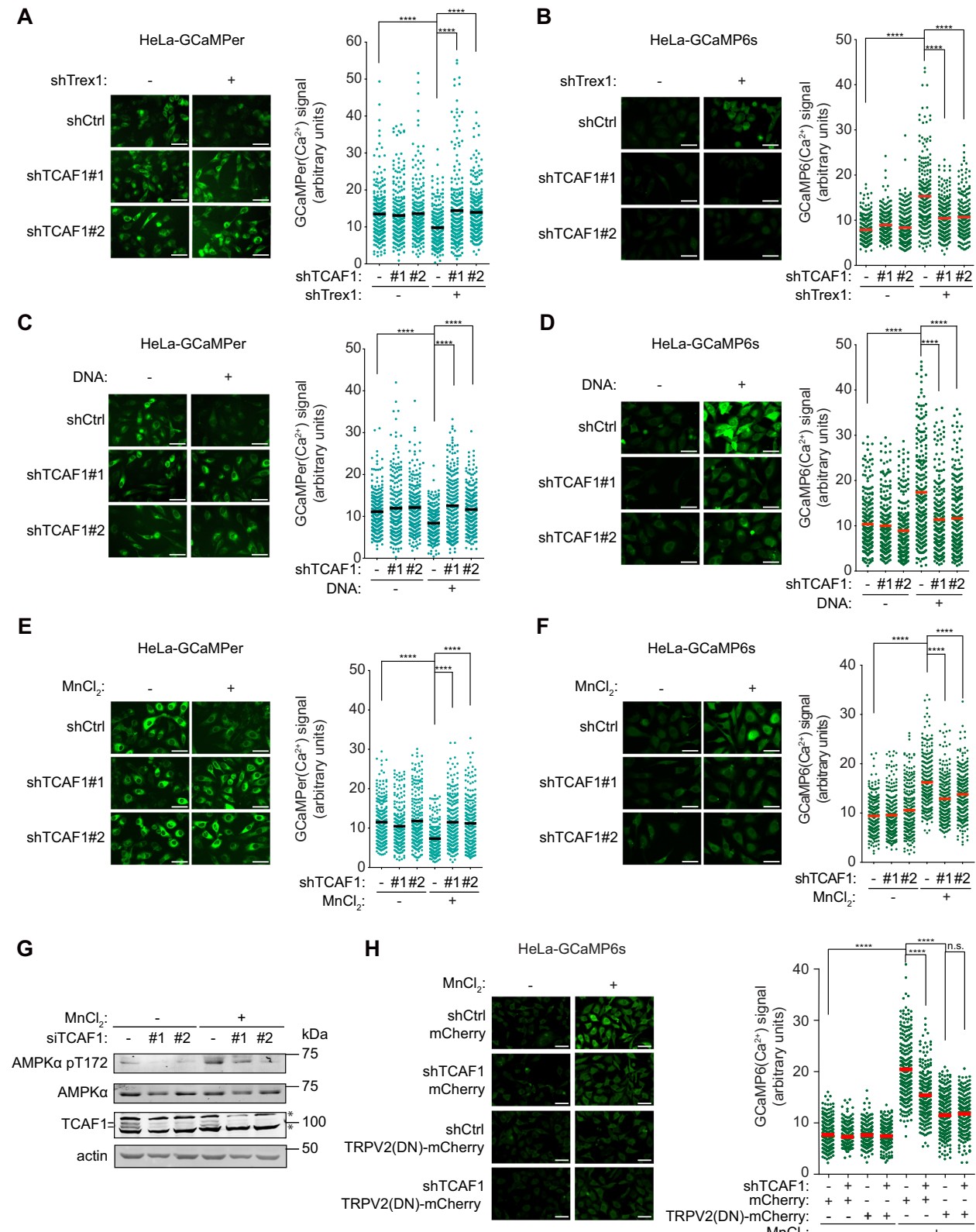

STING-mediated repression because its presence is no longer required for the Ca²⁺ release by TRPV2 when STING is depleted. Future work is required to pinpoint exactly how TCAF1 facilitates the dissociation between STING and TRPV2.

We observed that both endogenous TCAF1 and exogenously expressed protein exhibit two bands on the western blot, suggesting that it undergoes post-translational modification(s) in cells. The role of the modification(s) in TCAF1 function is unclear at this point, but we have not observed obvious alterations in the TCAF1 band pattern before and after replication stress or under conditions that cause cytosolic DNA or cGAS/STING activation (data not shown). It has been shown that the function of TCAF1 in regulating TRPM8 is antagonized

**Fig. 3 | TCAF1 is required for Ca²⁺ release in response to cytosolic DNA or direct cGAS activation. A** Effects of TCAF1 knockdown on ER Ca²⁺ release induced by Trex1 knockdown in HeLa cells. Left panel: representative images of GCaMPer signal (scale bar, 25 μm). Right panel: quantified GCaMPer signals in the samples depicted in the left panel. 250 cells were scored for the GCaMPer signal in each sample. Black bars represent the median. $n = 3$, ****, $p \leq 0.0001$ (two-tailed, unpaired t-test). See the source data for the exact $P$ values. **B** Effects of TCAF1 knockdown on iCa²⁺ elevation induced by Trex1 knockdown in HeLa cells. Left panel: representative images of GCaMP6s signal (scale bar, 25 μm). Right panel: quantified GCaMP6s signals in the samples depicted in the left panel. 250 cells were scored for each sample. Red bars represent the median. $n = 3$, ****, $p \leq 0.0001$ (two-tailed, unpaired t-test). See the source data for the exact $P$ values. **C** Effects of TCAF1 knockdown on ER Ca²⁺ release induced by DNA transfection in HeLa cells. HeLa cells were transfected with plasmid DNA (2 μg/ml), and the GCaMPer signal was imaged 7 h after transfection. Left panel: representative images of GCaMPer signal (scale bar, 25 μm). Right panel: quantified GCaMPer signals in the samples depicted in the left panel. 250 cells were scored for each sample. Black bars represent the mean. $n = 3$, ****, $p \leq 0.0001$ (two-tailed, unpaired t-test). See the source data for the exact $P$ values. **D** Effects of TCAF1 knockdown on iCa²⁺ elevation induced by plasmid DNA transfection (2 μg/ml, 7 h) in HeLa cells. Left panel: representative images of GCaMP6s signal (scale bar, 25 μm). Right panel: quantified GCaMP6s signals in the samples depicted in the left panel. 250 cells were scored for each sample. Black bars represent the mean. $n = 3$, ****, $p \leq 0.0001$ (two-tailed, unpaired t-test). See the source data for the exact $P$ values. **E** Effects of TCAF1 knockdown on ER Ca²⁺ release induced by MnCl₂ treatment (0.5 mM, 1.5 h) in HeLa cells. Left panel: representative images of GCaMPer signal (scale bar, 25 μm). Right panel: quantified GCaMPer signals in the samples depicted in the left panel. 250 cells were scored for each sample. Black bars represent the mean. $n = 3$, ****, $p \leq 0.0001$ (two-tailed, unpaired t-test). See the source data for the exact $P$ values. Outlier signals were removed through ROUT (Q = 1%) analysis. **F** Effects of TCAF1 knockdown on iCa²⁺ elevation induced by MnCl₂ treatment (0.5 mM, 1.5 h) in HeLa cells. Right panel: representative images of GCaMP6s signal (scale bar, 25 μm). Right panel: quantified GCaMP6s signals in the samples depicted in the left panel. 250 cells were scored for each sample. Red bars represent the mean. $n = 3$, ****, $p \leq 0.0001$ (two-tailed, unpaired t-test). See the source data for the exact $P$ values. Outlier signals were removed through ROUT (Q = 1%) analysis. **G** Effects of TCAF1 knockdown on AMPKα-phosphorylation at T172 in HeLa cells after MnCl₂ treatment (2.5 mM, 4 h). **H** Effect of TRPV2(DN) expression on iCa²⁺ elevation in control-knockdown and TCAF1-knockdown HeLa cells after MnCl₂ treatment (0.5 mM, 1.5 h). Left panel: representative images of GCaMP6s signal (scale bar, 25 μm). Right panel: quantified GCaMP6s signals in the samples depicted in the left panel. 250 cells were scored for each sample. Red bars represent the mean. $n = 3$, ****, $p \leq 0.0001$ (two-tailed, unpaired t-test). See the source data for the exact $P$ values. Source data are provided as a Source Data file.

by its related protein TCAF2[29,47]. It will be interesting to determine if TCAF2 also counteracts TCAF1 in the STING-TRPV2 dissociation and subsequent TRPV2-mediated Ca²⁺ release in response to cytosolic DNA. Further investigation of TCAF1's functions in TRPV2 regulation, replication fork protection, and its functional relationships with STING and TCAF2 will enhance our understanding of the mechanisms of genome maintenance and Ca²⁺ regulation in response to cytosolic DNA and may unveil novel therapeutic targets for cancer and other diseases.

## Methods

### TCAF1 antibody generation and purification
Rabbit antibodies against human TCAF1 were raised using GST-TCAF1(1-279) protein expressed in and purified from E. coli. To affinity-purify the antibodies, the GST-TCAF1(1-279) antigen was transferred onto the PVDF membrane followed by Ponceau S staining for protein visualization. The excised PVDF slices with protein antigen were then incubated with antisera overnight at 4 °C. Antibodies were then eluted with 0.1 M glycine (pH 2.5) and neutralized by the addition of 0.1 volume of 1 M Tris-HCl (pH 8.4), followed by concentration using an Ultra-4 centrifugal filter unit (Millipore).

### Cell culture, transfection, and generation of cell lines with stable gene expression or knockdown
HeLa, U2OS, and HEK293T cells were cultured in Dulbecco's Modified Eagle's Medium (DMEM) supplemented with 10% fetal bovine serum (FBS), 100 U/ml penicillin and 100 μg/ml streptomycin at 37 °C with 5% CO₂ in a humidified incubator. Non-transformed MCF10A cells were cultured in DMEM/F-12 supplemented with 5% horse serum, 20 ng/ml EGF, 0.5 mg/ml hydrocortisone, 100 μg/ml cholera toxin, 10 μg/ml Insulin, 100 U/ml penicillin and 100 μg/ml also at 37 °C with 5% CO₂ in a humidified incubator. Plasmids were transfected into cells using TransIT-LT1 (Mirus, HEK293T) or Lipofectamine 3000 (Invitrogen, for HeLa) according to the protocols of the manufacturers. siRNA transfection was done using Lipofectamine RNAiMAX transfection reagent (Invitrogen) according to the protocol of the manufacturer. The plasmids used in this study are listed in Supplementary Data 3.

shRNA-mediated knockdown and over-expression of genes were done through lentiviral transduction. GCaMP6s- and GCaMPer-expressing lentiviruses were generated in HEK293T cells through co-transfection of pMDLg/pRRE, pRSV.Rev and pMD2.G with pBOB-GCaMP6s or pBOB-GCaMPer. The packaging plasmids psPAX2 and pMD2.G were used to produce all the other lentiviruses used in this study. Viral supernatant was collected 48 h and 72 h after transfection and filtered using Millex-HV Syringe Filter (0.45 μm, Millipore Sigma). Target cells were transduced with filtered viral supernatant in the presence of polybrene (10 μg/ml). Cells stably expressing GCaMP6s or GCaMPer were obtained through cell sorting and single clones were screened by imaging and western blot. Cells infected with lentiviruses expressing shRNAs, TCAF1, STING, or TRPV2 were selected with puromycin (1.5 μg/ml) for 2 days. All the stable cell lines were used for experiments after at least 7 days post-infection to ensure that no cytosolic DNA was generated from lentiviral RNA through reverse transcription. The shRNA targeting sequences are listed in Supplementary Data 3.

### Generation of stable cell lines of gene knockout
To generate TCAF1-KO or Exo1-KO cell lines, pCRSIPRv2-sgRNA-expressing constructs with sgRNAs targeting human TCAF1 or Exo1 genes were transfected into HeLa cells. Twenty-four hours after transfection, cells were selected with puromycin (1.5 μg/ml) for 2 days. Single cells were grown in 96-well plates for amplification. Individual clones were verified by western blot to detect the depletion of TCAF1. The sgRNA targeting sequences are listed in Supplementary Data 3.

### Genome-wide CRISPR/Cas9 screen to identify factors that promote cell survival after replication stress
Wild type and Exo1-KO HeLa cells were infected with lentiviruses expressing Cas9 and then selected with blasticidin (10 μg/mL) for 5 days to establish the Cas9-expressing cell lines. To carry out the screen, HeLa-Cas9 cells were infected with lentiviruses expressing the GeCKOv2 library at a multiplicity of infection (MOI) of 0.3 with a 500x coverage of the library. Two days after infection, puromycin selection was done to eliminate uninfected cells. Eight days after puromycin selection, cells were treated with 4 mM HU for 24 hours and recovered in fresh DMEM medium for 4 days. Genomic DNA was extracted using PureLink Genomic DNA kit (ThermoFisher Scientific, K182001) followed by two PCR reactions to prepare samples for Illumina Next-Gen sequencing. The first PCR was used to amplify sgRNA inserts in the cells, and the second PCR was used to add Illumina sequencing tags as well as index tags for sample identification. To improve the complexity of the library required for deep sequencing, a mixture of 5 forward primers with staggered nucleotides immediately upstream of the sgRNA sequences was used in the second PCR. The sequences of

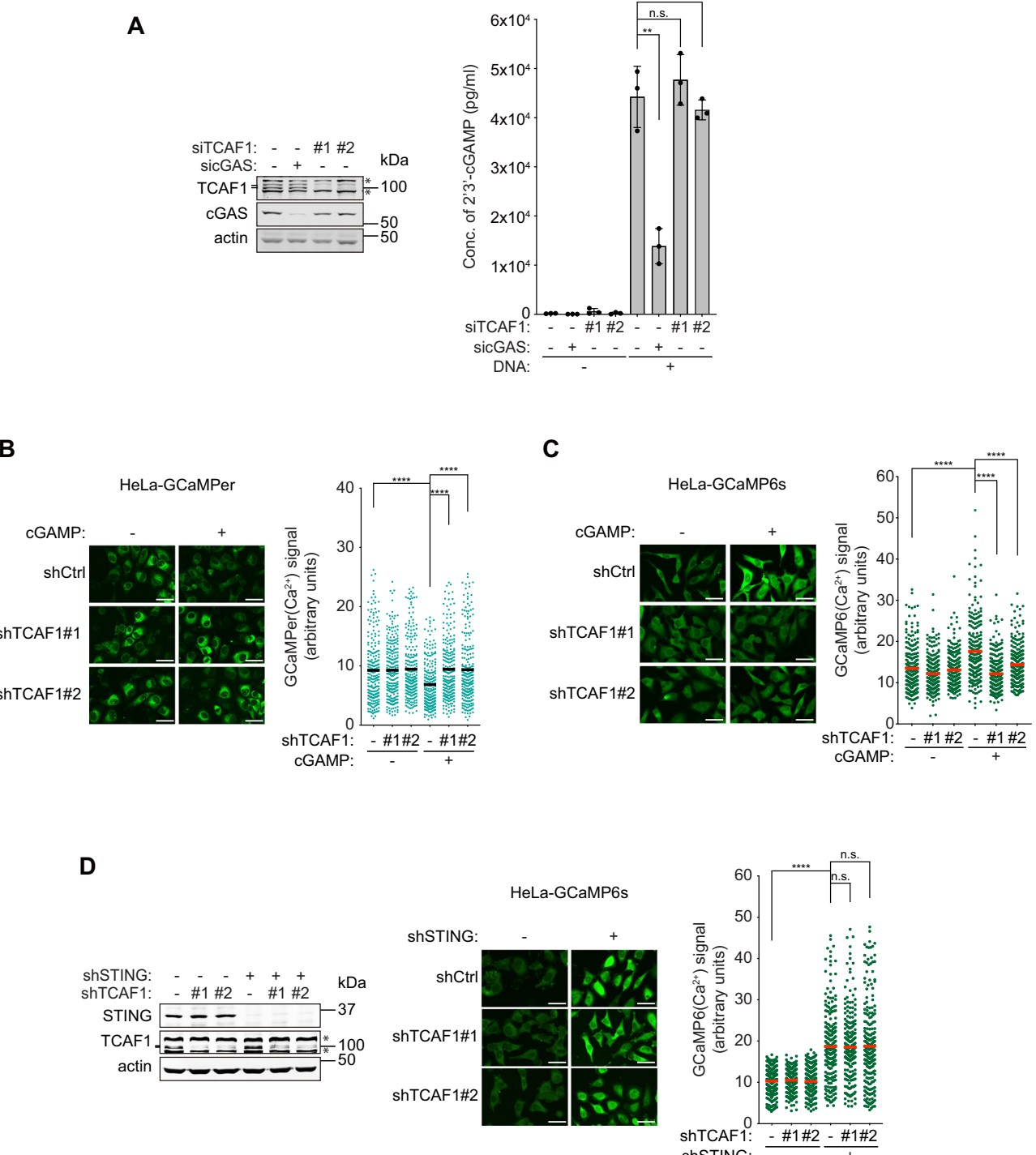

**Fig. 4 | TCAF1 promotes iCa²⁺ elevation downstream of cGAMP. A** Left panel: Western blot analysis siRNA-mediated knockdown of TCAF1 or cGAS in HeLa cells. *, nonspecific bands. Right panel: ELISA analysis of 2'3'-cGAMP concentration in HeLa cells transfected with plasmid DNA (2 µg/ml, 7 h). Data represent mean ± S.D. from triplicate. **, $p \le 0.01$ (two-tailed, unpaired t-test). See the source data for the exact *P* values. **B** Effects of TCAF1 knockdown on ER Ca²⁺ release induced by cGAMP in HeLa cells. Cells were transfected with cGAMP (5 µg/ml), and the GCaMPer signal was imaged 7 h after transfection. Left panel: representative images of GCaMPer signal (scale bar, 25 µm). Right panel: quantified GCaMPer signals in the samples depicted in the left panel. 250 cells were scored for each sample. Black bars represent the mean. $n = 3$, ****, $p \le 0.0001$ (two-tailed, unpaired t-test). See the source data for the exact *P* values. Outlier signals were removed through ROUT (Q = 1%) analysis. **C** Effects of TCAF1 knockdown on iCa²⁺ elevation induced by

cGAMP in HeLa cells. Cells were transfected with cGAMP (5 µg/ml), and the GCaMP6s signal was imaged 7 h after transfection. Left panel: representative images of GCaMP6s signal (scale bar, 25 µm). Right panel: quantified GCaMP6s signals in the samples depicted in the left panel. 250 cells were scored for each sample. Red bars represent the mean. $n = 3$, ****, $p \le 0.0001$ (two-tailed, unpaired t-test). See the source data for the exact *P* values. **D** Effects of TCAF1 knockdown on iCa²⁺ elevation induced by STING depletion in HeLa cells. Left panel: Western blot analysis shows the depletion of TCAF1 and STING in HeLa-GCaMP6s cells. Middle panel: representative images of GCaMP6s signal (scale bar, 25 µm). Right panel: quantified GCaMP6s signals in the samples depicted in the middle panel. 250 cells were scored for GCaMP6s signal in each sample. Red bars represent the mean. $n = 3$, ****, $p \le 0.0001$ (two-tailed, unpaired t-test). See the source data for the exact *P* values. Source data are provided as a Source Data file.

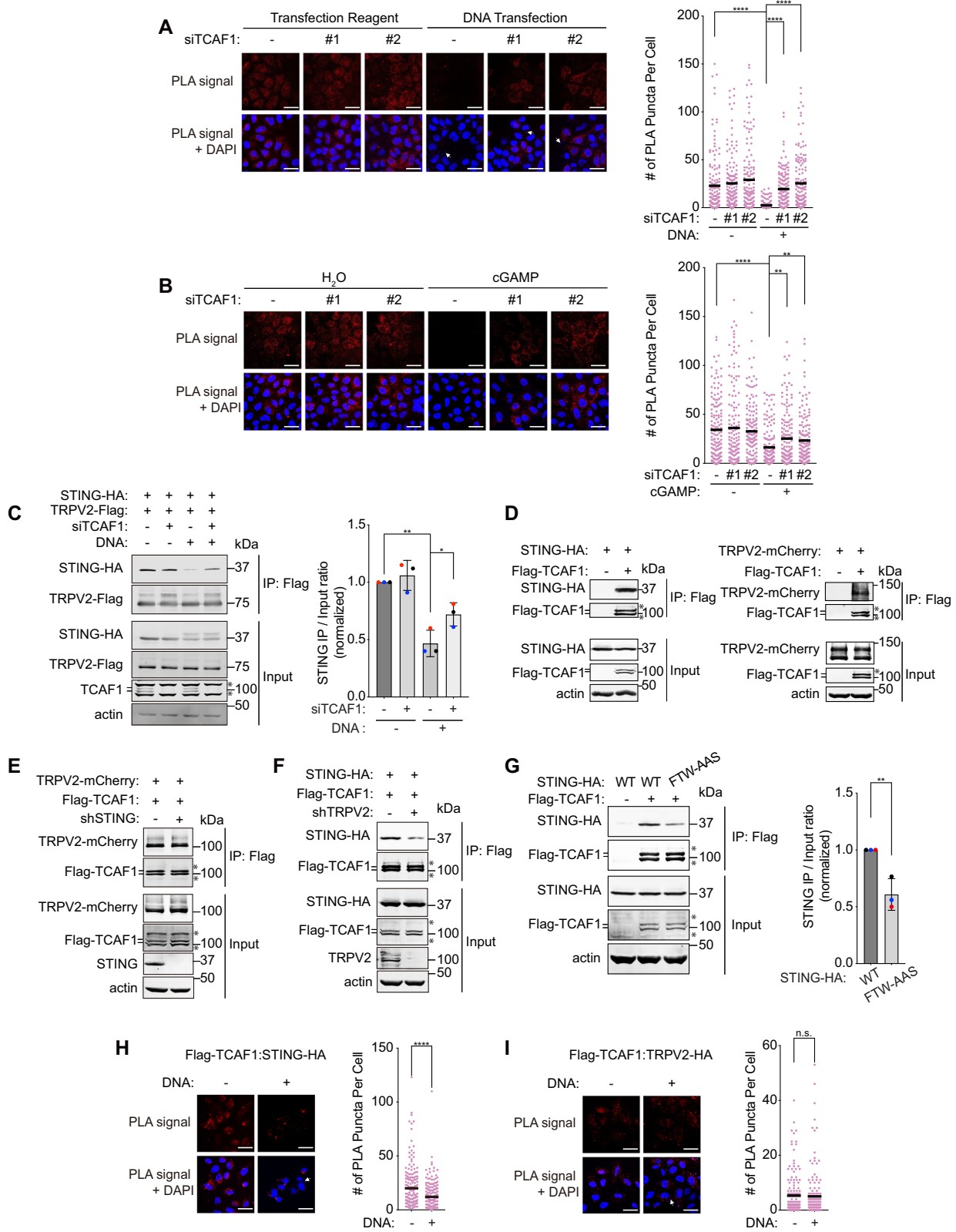

primers used are listed in Supplementary Data 3. All PCR reactions were performed using Phusion Hot Start II High-Fidelity DNA Polymerase (ThermoFisher Scientific, F549L). PCR samples were sequenced using the Illumina HiSeq 2500 platform[48]. A custom Perl script was used to determine the read counts for each sgRNA and map the sgRNAs to their gene IDs in the reference GeCKOv2 library. The

script is available upon request. Only the sgRNAs with at least 20 reads in each sample were used for further analysis. The reads numbers for three TCAF1 sgRNAs were 816, 52, and 1009 in untreated samples and decreased to 49, 37, and 27 in HU-treated samples. Analysis of genes essential for cell survival upon HU treatment was performed using MAGeCK, a computational tool designed to rank genes based on the

**Fig. 5 | TCAF1 promotes the dissociation of STING from TRPV2 in response to cytosolic DNA or direct STING activation. A** Left panel: Representative images of PLA signal of TRPV2-Flag and STING-HA in control-knockdown or TCAF1-knockdown HeLa cells transfected with plasmid DNA (2 μg/ml, 7 h) (scale bar, 25 μm), transfected DNA are marked by arrows. Right panel: Quantified PLA signal of 150 cells of the samples depicted in the left panel. Black bars represent the mean. $n = 3$, ****, $p ≤ 0.0001$ (two-tailed, unpaired t-test). See the source data for the exact $P$ values. **B** Left panel: Representative images of PLA signal of TRPV2-Flag and STING-HA in control-knockdown or TCAF1-knockdown HeLa cells transfected with cGAMP (5 μg/ml, 7 h) (scale bar, 25 μm). Right panel: Quantified PLA signal of 150 cells of the samples depicted in the left panel. Black bars represent the mean. $n = 3$, ****, $p ≤ 0.0001$. **, $p ≤ 0.01$ (two-tailed, unpaired t-test). See the source data for the exact $P$ values. **C** Effects of TCAF1 knockdown on the dissociation of STING from TRPV2 induced by cytosolic DNA. Left panel: Representative co-IP result for TRPV2-Flag and STING-HA in control- or TCAF1-knockdown HeLa cells after plasmid DNA transfection (2 μg/ml, 7 h). *, nonspecific bands. Right panel: Quantified IP/Input ratios for STING-HA. Data represent mean ± S.D., $n = 3$ **, $p ≤ 0.01$. *, $p ≤ 0.05$ (two-tailed, unpaired t-test). See the source data for the exact $P$ values. **D** Result of co-IP for Flag-TCAF1 and STING-HA (left panel) and for Flag-TCAF1 and TRPV2-mCherry (right panel) in 293 T cells. **E** Result of co-IP on the effects of STING knockdown on

the association between TCAF1 and TRPV2 in 293 T cells. **F** Result of co-IP on the effects of TRPV2 knockdown on the association between TCAF1 and STING in 293 T cells. **G** Left panel: Representative co-IP result for the association of Flag-TCAF1 with STING(WT)-HA or STING(FTW/AAS)-HA in 293 T cells. Right Panel: Quantified IP/Input ratios for STING-HA from three independent experiments. Data represent mean ± S.D., $n = 3$, **, $p ≤ 0.01$ (two-tailed, unpaired t-test). See the source data for the exact $P$ values. **H** Effects of DNA transfection on the association between Flag-TCAF1 and STING-HA. Left panel: Representative images of PLA signal of Flag-TCAF1 and STING-HA in HeLa cells transfected with or without plasmid DNA (2 μg/ml, 7 h) (scale bar, 25 μm), transfected DNA are marked by arrows. Right panel: Quantified PLA signal of 150 cells of the samples depicted in the left panel. Black bars represent the mean. $n = 3$, ****, $p ≤ 0.0001$ (two-tailed, unpaired t-test). See the source data for the exact $P$ values. **I** Effects of DNA transfection on the association between Flag-TCAF1 and TRPV2-HA. Left panel: Representative images of PLA signal of Flag-TCAF1 and TRPV2-HA in HeLa cells transfected with plasmid DNA (2 μg/ml, 7 h) (scale bar, 25 μm), transfected DNA are marked by arrows. Right panel: Quantified PLA signal of 150 cells of the samples depicted in the left panel. Black bars represent the mean. $n = 3$. n.s., not significant. See the source data for the exact $P$ values.

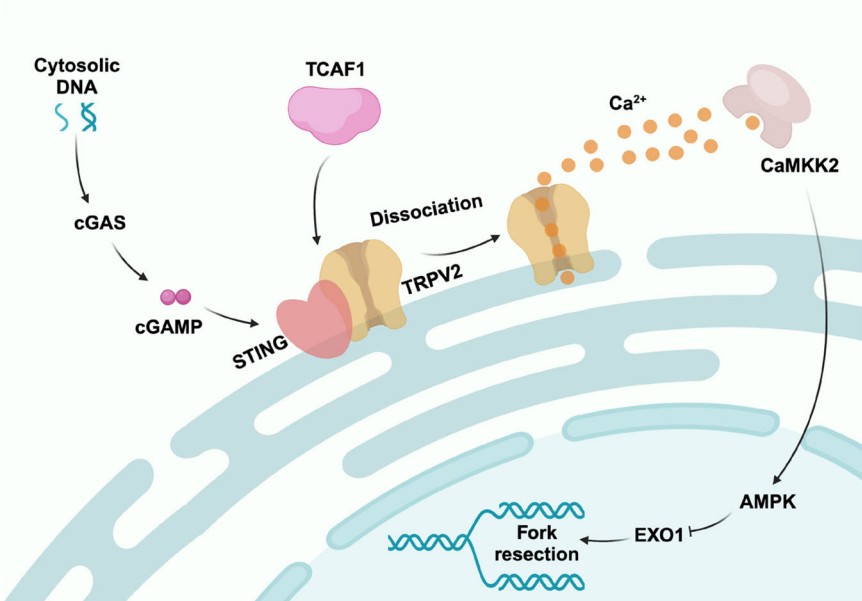

**Fig. 6 | TCAF1 promotes fork protection by facilitating STING-TRPV2 dissociation and subsequent Ca²⁺ release in response to cytosolic DNA induced by replication stress.** A model for the role of TCAF1 in the cytosolic DNA-triggered, TRPV2/Ca²⁺-dependent replication fork protection pathway (see text for details). Figure 6 was created with BioRender.com released under a Creative Commons Attribution-NonCommercial-NoDerivs 4.0 International license.

enrichment/depletion of individual sgRNAs as well as the number of enriched/depleted sgRNAs for each gene[31].

## Nondenaturing BrdU immunofluorescence staining for measuring fork resection

BrdU incorporation followed by nondenaturing BrdU immunofluorescence staining for measuring replication fork resection was described previously[19,20]. Briefly, cells cultured on glass-bottomed dishes were incubated with BrdU (10 μM) for 36 h and then treated with HU (2 mM, 5 h) to induce replication stress. Cells were then permeabilized with freshly made extraction buffer (10 mM PIPES pH 6.8, 100 mM NaCl, 300 mM sucrose, 3 mM MgCl₂, 1 mM EGTA and 0.2% Triton X-100) for 5 min. Subsequently, cells were fixed with 3% paraformaldehyde in phosphate-buffered saline (PBS) for 20 min at room temperature, cold methanol (−20 °C) for 20 min, and then ice-cold acetone (4 °C) for 30 seconds. Subsequently, cells were blocked with blocking buffer (PBS containing 0.05% Tween-20 and 2% bovine serum albumin (BSA)) for 1 h at room temperature, followed by

immunostaining with anti-BrdU antibodies (1:1000, BD Pharmingen, 555627) overnight at 4 °C. Cells were then incubated with goat anti-mouse secondary antibody (1:500, Invitrogen, A11001) for 1 h at room temperature. Both primary and secondary antibodies were diluted in PBS containing 2% BSA. After nuclear staining with Hoechst 33342 (1 μg/ml), images were captured using an inverted microscope (Nikon Ti-E) and Metamorph software (Molecular Devices). The BrdU signal in individual nuclei (defined by the Hoechst-stained area) was determined using ImageJ. Cells with a BrdU signal above that in the majority (98%) of untreated control cells were taken as BrdU-positive. Images of at least 1,000 randomly selected cells for each sample were quantified. Statistical analysis was performed in GraphPad Prism using an unpaired t-test.

## Single-molecular DNA fiber assay for measuring fork resection

Single-molecular DNA fiber assay for measuring the resection of nascent DNA at replication forks was described previously[49,50]. HeLa cells were pulse-labeled sequentially with prewarmed fresh media

containing thymidine analogs iododeoxyuridine (IdU, 20 μM) and chlorodeoxyuridine (CldU, 200 μM) sequentially for 20 min each. Cells were then washed with PBS and treated with HU (4 mM) for 2 h. After this, cells were then trypsinized and resuspended in ice-cold PBS to a final concentration of 500,000 cells/ml. 2 μl of cell suspension was spotted onto one edge of a precleaned glass slide and set for 1 min, then 8 μl of spreading buffer (200 mM Tris-HCl pH7.5, 50 mM EDTA, 0.5 % SDS) was added in drops onto the cells for lysis. After 6 min of incubation, the slides were tilted (20–45°) to allow the liquid to slowly run down the length of the slide to spread the genomic DNA. Next, slides were air dried, fixed in pre-cold methanol-acetic acid (3:1) for 10 min, and then denatured with 2.5 N HCl for 1 h at room temperature. After rinsing with PBS, the slides were blocked in 5% BSA in PBS with 0.1% Tween-20 for 1 h at room temperature. Immunodetection of DNA fibers was performed with rat anti-BrdU antibody (1:50, Abcam, ab6326) for CldU and mouse anti-BrdU antibody (1:50, Becton Dickson, 347580) for IdU in a humidity chamber at 37 °C for 2 h. Slides were then incubated with secondary antibodies (anti-rat Alexa 488 (Molecular Probes, A21470, 1:100) and anti-mouse Alexa 546 (Molecular Probes, A21123, 1:100)) at 37 °C for 45 min in the dark. After washing with PBST (0.1% Tween 20), excess liquid was drained from the slides followed by mounting with Prolong Gold Antifade (ThermoFisher Scientific). Images of the DNA fibers were captured by using a 60× oil immersion objective of an inverted fluorescence microscope (Nikon Ti-E microscope). 150 fiber tracts were scored for each sample. The DNA track lengths were measured using ImageJ and the pixel length values were converted into micrometers using the scale bars generated by the microscope.

## Metaphase chromosome spreading assay for measuring chromosomal aberrations
Chromosomal aberrations were detected in DAPI-stained metaphase spreads, as described previously[20,34]. HeLa cells treated with 4 mM HU for 6 h were released in fresh medium to recover for 20 h. Cells were then treated with 10 μM nocodazole for 4 h to induce cell cycle arrest before harvest. Trypsinized cells were then resuspended in 10 ml of pre-warmed hypotonic solution (10 mM KCl and 10% FBS) for 10 minutes at 37 °C, followed by fixation and in ice-cold fixation buffer (1: 3 acetic acid: methanol) for 30 minutes on ice. Cells were next washed with ice-cold fixation buffer for 4 times and dropped onto pre-chilled slides to obtain chromosome spreads. The slides were air-dried thoroughly and mounted with Prolong Gold Antifade reagent with Hoechst. Images were captured by using a 60× oil immersion objective of an inverted fluorescence microscope (Nikon Ti-E microscope). 150 randomly selected metaphases were scored per sample.

## Clonogenic assay for measuring cell viability
HeLa cells were plated on 6-well dishes at a seeding density of 400 cells per well. 16 hours after plating, cells were treated with HU or bleomycin at the indicated concentrations for 24 h, or with camptothecin (CPT) at the indicated concentrations for 4 h. Cells were then washed with PBS and cultured in fresh medium for 10 days to allow colony formation. Colonies were then stained with 0.2% Crystal Violet in 50% methanol and washed in water before being counted in ImageJ.

## BrdU incorporation and cell cycle analysis
Cell cycle and DNA replication analysis were performed by flow cytometry after BrdU incorporation, as described before[19]. Cells were pulsed with BrdU (20 μM) for 30 min, and then trypsinized. After washing with PBS, cells were fixed in 70% ethanol at −20 °C overnight. Subsequently, cells were pelleted down and then incubated with 2 N HCl/0.5% Triton X-100 for 30 min at room temperature to denature DNA followed by neutralization in 0.1 M sodium tetraborate (pH 8.5). Cells were next incubated with mouse anti-BrdU antibody (1:400, BD Biosciences, 347580) in antibody dilution buffer (PBS + 0.5% Tween

20 + 1% BSA) overnight at 4 °C. After incubation, cells were washed 3 times with PBS containing 1% BSA and then incubated with Alexa Fluor 488-conjugated goat anti-mouse IgG (1:500, Thermofisher, A-11001) for 1 h. After washing with PBS containing 1% BSA, cells were resuspended in PBS containing propidium iodide (20 μg/ml) and RNase A (200 μg/ml) and incubated at 37 °C for 30 min in the dark. Flow cytometry was performed on a BD FACScan5 Flow Cytometer, and the cell cycle profile was analyzed with FlowJo software.

## Live cell imaging of Ca²⁺ using the GCaMP6s and GCaMPer reporters
For $Ca^{2+}$ imaging in live cells, genetically encoded calcium indicators GCaMP6s and GCaMPer were used to measure $Ca^{2+}$ levels in the cytoplasm and ER, respectively, as described before[39,40]. To measure the intracellular $Ca^{2+}$ elevation or ER $Ca^{2+}$ release after HU treatment, cells expressing GCaMP6s or GCaMPer were first synchronized in the early S phase through a double-thymidine block procedure. Briefly, cells cultured on 35 mm glass-bottomed dishes were first treated with 2 mM thymidine for 18 h, followed by a 9 h release in fresh medium, and then treated again with 2 mM thymidine for 17 h before release into fresh medium for 2 h. Cells were then treated with HU (4 mM, 4 h) to induce replication stress before imaging. $Ca^{2+}$ imaging was also performed for asynchronized cells with different treatments as indicated in figure legends. Fluorescence signals of GCaMP6s or GCaMPer were acquired in a live-cell imaging chamber capable of maintaining temperature, humidity, and $CO_2$ levels, using an inverted Nikon Ti-E microscope with an objective of 20×. 250 cells were scored for each sample. Fluorescence signals in individual cells were quantified using ImageJ after subtracting the background.

## Detection of 2'3'-cGAMP in cells using ELISA
cGAMP ELISA was performed according to the manufacturer's protocol using a 2'3'-cGAMP ELISA Kit (Cayman Chemical, 501700). For sample preparation, HeLa cells were lysed through a freeze and thaw cycle in liquid nitrogen (30 sec) and a 37 °C water bath (5 min) for 3 times. Cells were then sonicated in iced water for 1 min and centrifuged at 13,000 g for 10 min at 4 °C to remove debris. The supernatant was used for cGAMP ELISA.

## Proximity ligation assay (PLA)
A proximity ligation assay (PLA) was performed in HeLa cells stably expressing TRPV2-Flag and STING-HA. Cells were seeded onto Lab-Tek II CC2 chamber slides (MilliporeSigma, S6815) and treated as indicated in the figure legends. After treatment, cells were fixed with 4% paraformaldehyde in PBS for 15 min and then permeabilized with 0.2% Triton X-100 in PBS for 15 min. After blocking with the Blocking Solution (MilliporeSigma, DUO82007) for 1 hour at 37 °C, cells were incubated with primary antibodies in the Antibody Diluent (MilliporeSigma, DUO82008) at 1:1,000 overnight at 4 °C. Cells were then washed in Wash Buffer A (MilliporeSigma, DUO82046) for 10 min at room temperature and incubated with PLUS and MINUS PLA probes (MilliporeSigma, DUO82002, DUO82004) at 1:5 in the Antibody Dilute for 1 hour at 37 °C. Cells were again washed in Wash Buffer A for 10 min at room temperature and then incubated with the Ligase (MilliporeSigma, DUO82027) in the Ligation buffer (MilliporeSigma, DUO82009) for 30 min at 37 °C, as described by the manufacturer. Subsequently, cells were washed in Wash Buffer A for 10 min at room temperature and incubated with the Polymerase (MilliporeSigma, DUO82028) in the Amplification Buffer (MilliporeSigma, DUO82011) for 100 min at 37 °C. After that, cells were washed in Wash Buffer B (MilliporeSigma, DUO82048) for 20 min at room temperature and then stained for 10 min in Wash Buffer B containing Hoechst 33342 (5 μg/ml). After washing in 0.01× Wash Buffer B for 1 min at room temperature cells were mounted with Prolong Gold Antifade reagent (Thermo Fisher, P36930). Fluorescent images were acquired using an

inverted Nikon Ti-E microscope with a 60×oil immersion objective. 150 cells were quantified for each sample.

## Immunofluorescence staining, immunoprecipitation, and immunoblotting

Immunofluorescent staining was performed as previously described[20]. Cells cultured on glass-bottomed dishes (Mattek, P35G-1.5-14-C) were washed with PBS and fixed with 4% paraformaldehyde for 10 min at room temperature. After washing 2 times with PBS, cells were permeabilized with PBS containing 0.2% Triton X-100 for 10 min and then blocked with PBS containing 10% goat serum for 1 h. Cells were incubated with primary antibodies (in PBS with 0.1% Triton X-100 and 10% goat serum) overnight at 4 °C and then with secondary antibodies (in PBS with 0.1% Triton X-100 and 10% goat serum) for 1 h at room temperature, followed by Hoechst 33342 (1 μg/ml) staining for 10 min. To detect AMPKα T172-phosphorylation, cells were first rinsed twice with PBS containing 0.1% Triton X-100 for pre-permeabilization to remove the signal in the cytoplasm, followed by the same immunofluorescence staining procedure described above. Fluorescence images were captured using an inverted Nikon Ti-E microscope.

To immunoprecipitate (IP) Flag-tagged proteins, cells were lysed in the lysis buffer (10 mM NaKPO4, pH 7.2, 150 mM NaCl, 1% sodium deoxycholate, 1% Triton X-100, protease inhibitor cocktail, and phosphatase inhibitor cocktail) and then sonicated in ice water for 40 sec[29]. The cell lysate was then centrifuged at 13,000 g for 10 min at 4 °C. Subsequently, the supernatant was incubated with 20 μl of anti-Flag M2 Magnetic Beads (Sigma, M8823) for 1.5 h at 4 °C. After washing 5 times with lysis buffer, bead-bound proteins were dissolved in SDS sample buffer and heated (37 °C, 30 min for TRPV2, and 95 °C, 10 min for all other proteins) before gel loading.

For immunoblotting, protein samples were resolved by SDS-PAGE and transferred to Immobilon®-P PVDF transfer membrane (Millipore, IPVH20200). Membranes were blocked with Casein blocking buffer (1×TBS with 0.1% Casein) and then incubated with the primary antibodies. After washing 3 times with wash buffer (1×TBS with 0.1% Tween-20), blots were incubated with DyLight 800- and DyLight 680-conjugated secondary antibodies followed by washing 3 times with wash buffer. Blots were then scanned by an Odyssey Imaging System (LI-COR Biosciences), as previously described[51]. For an example of the presentation of full scan blots, see the Source Data file.

## Quantification and statistical analysis

Statistical significance tests were done with the GraphPad Prism software. The sample size (n), the number of independent replicates for each experiment, and the tests performed are depicted in the figure legends.

## Reporting summary

Further information on research design is available in the Nature Portfolio Reporting Summary linked to this article.

## Data availability

The raw Genome-wide CRISPR/Cas9 screen sequencing data generated in this study have been deposited in NCBO's Gene Expression Omnibus (GEO) database with the accession code GSE244205. The processed sequencing data are provided as Supplementary Data 1 and 2. Source data including original Western blot and microscopic signal quantifications are provided with this paper. Source data are provided with this paper.

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

## Acknowledgements

We are grateful to Dr. Dimitra Gkika for the gift of the TCAF1 expression construct. We thank all members of the You lab for their technical assistance and critical discussions. This work was supported by NIH grants (R01GM098535 to Z.Y. and R01CA193318 to N.M.), an American Cancer Society Research Scholar Grant (RSG-13-212-01-DMC to Z.Y.), a Siteman Investment Program grant from Washington University (5124 to Z.Y.), and the National Natural Science Foundation of China (82272984 to S.L.).

## Author contributions

Z.Y. and S.L. conceived and supervised the project. L.K., S.L. and C.C. conducted the experiments and analyzed the results with contributions from A.C., J. Y., Y.Y., S.H., D. F., N.T., E.W. under the supervision of Z.Y., N.M., M.B.M. and D.W.P. All authors read and approved the manuscript.

## Competing interests

The authors declare no competing interests.
