## [Peer Review File · Nature Communications]

TCAF1 Promotes TRPV2-mediated Ca²⁺ Release in Response to Cytosolic DNA to Protect Stressed Replication ForksREVIEWER COMMENTS

Reviewer #1 (Remarks to the Author):

In this manuscript, Kong et al identifies TCAF1 as a key factor promoting TRPV2-mediated Ca²⁺ release under conditions that activate cGAS/STING, highlighting the mechanism by which cytosolic DNA activates TRPV2. TCAF1 achieves this by facilitating the dissociation of STING from TRPV2, relieving TRPV2 repression. The authors further demonstrated that the role of TCAF1 is crucial for fork protection, chromosomal stability, and cell survival during replication stress. The manuscript is clearly written, the figures are presented in a logic way. Understanding how the TRPV2-mediated Ca²⁺ release pathway is activated by cytosolic DNA through the involvement of TCAF1 provides valuable insights into the molecular basis of genome maintenance and cancer prevention. This study thereby has the potential to impact our understanding of cancer biology and may contribute to the development of new strategies for cancer prevention and treatment. However, to further strengthen the molecular mechanisms presented in this study, additional evidence and explorations are warranted. For instance, it is crucial to investigate whether TCAF1 exhibits sensitivities to other types of DNA damage reagents, beyond HU. Moreover, exploring whether phenotypes observed in TCAF1 KD cells can be rescued by other replication or repair enzymes, besides EXO1, would provide a more comprehensive understanding. The most importantly, it is unclear what function of TCAF1 contributes to TRPV2 dissociation from STING. Mutants of TCAF1 which could not release TRPV2 should be identified for understanding the molecular mechanisms. Addressing these additional mechanistic insights and some methodological considerations will further strengthen its impact and contribution to the field.

Figure 1: Figure 1B raises questions regarding the sensitivity of TCAF1 to other forms of stress and DNA damage. It is essential to ascertain whether TCAF1 KD specifically sensitizes cells to replication stress or if it exhibits sensitivity to other DNA damage reagents in addition to HU

Figure 2 shows phenotypes observed in TCAF1 KD cells can be rescued in EXO1 KD cells. It would be valuable to explore whether other fork protection enzymes can also rescue some of the phenotypes of TCAF1 KD cells.

Figure 3 demonstrates that TCAF1 promotes TRPV2-mediated Ca²⁺ release after cGAS activation in response to cytosolic DNA utilizing TREX1 KD or plasmid transfection. However, there is a need to establish a connection between this phenomenon and replication fork protection, as cytosolic DNA can be induced by various other factors. It is essential to confirm whether this is a replication-specific reaction or a general response triggered by accumulated DNA damage.

Figure 5. Separation TCAF1-TRPV2 association and TCAF1-STING interaction will be important to prove the function of TCAF1. Elucidating the interaction between TCAF1 and different partners through truncations and mutants, coupled with structural analysis, would be instrumental in understanding how TCAF1 interacts with these partners and contributes to its function.

Other issues:

The authors described CRISPR KO of TCAF1 and EXO1 and clone selection. In the manuscript, all cell lines utilized siRNA mediated KD but not KO. It is important to address certain methodological inconsistencies, such as the use of siRNA-mediated KD rather than KO for cell lines in the manuscript.

The authors should provide evidence of the interaction between STING(FTW/AAS) and TRPV2 through immunoprecipitation experiments in Figure 5F.

Reviewer #2 (Remarks to the Author):

In this manuscript, You and colleague argue that TCAF1 promotes TRPV2-mediated Ca²⁺ upon the activation of cGAS-STING pathway. The authors use KO/KD methods then assess the effect of TCAF1. Although it is commendable that the authors used an array of measurements the effect is rather marginal. Below I list major issues:

- Authors have also generated knockout TCAF1 HeLa cells but still decided to use silenced TCAF1 cells throughout the manuscript. Why not use TCAF1 KO HeLa cells instead?
- Although the effect of TCAF KO/KD is statistically significant, I am fully convinced with the significance - the difference is minor (often $\leq 20\%$ and heavily eschewed by outliers) in all the experiments presented in this manuscript.
- In Fig.1C and other quantitative analyses presented, it seems that the mean is shifted because of the outlier and there seems to be no difference between before and after silencing of TCAF1.
- Why did authors decide to use short hairpin technique in some experiments and silencing RNA in other in this manuscript?
- Authors should also include the Western blots for AMPK phosphorylation in Fig. 2F like in ChK1 phosphorylation.
- Fig.S3A, fluorescence signal is very weak in shCtrl-shTrex1 which was not decreased further after silencing Trex1. This explains the poor difference in quantification results.
- Authors should move the fluorescence images of Fig.S3 to Fig.3 to make it clearer.
- Again in Fig.3D, the elevation is so minor that it seems like it's been caused by the outlier in quantification.
- Fig.3F, S3F, background signal is high after MnCl₂ treatment which is making it difficult to believe that after silencing TCAF1, phenotype is only partially rescued. Authors should keep similar baseline as in other figures.
- Again, as is throughout this manuscript, in Fig.4B, in the first lane vs 4th lane, shift in mean seems to be caused by the outliers in the 1st lane and there seems to be no difference if you remove those outliers.
- Elevation of iCa²⁺ after STING depletion is quite weak in Fig S4C
- Authors should move fluorescence images from Fig.S4 to Fig.4
- PLA signal images are missing from Fig.5E and PLA result after MnCl₂ treatment is not shown contrary to being mentioned in the text.
- Actin loading control is missing from Fig.5F and one should not see any STING being pulled down in lane 1 without TCAF.
- No mention of Fig.S5 anywhere in the text.

Reviewer #3 (Remarks to the Author):

In this manuscript, the authors identified TCAF1 as a new player in the cGAS-STING-TRPV2-Ca²⁺-CAMMK2-AMPK-EXO1 pathway previously discovered by their lab. They showed that TCAF1 is required for TRPV2-mediated Ca²⁺ release by promoting STING-TRPV2 dissociation upon STING activation, supporting the idea that TCAF1 is a new node in this signaling pathway. The experiments were nicely done, and the conclusions are essentially supported by the data shown. However, this reviewer has a few minor concerns.

1. TCAF1 apparently is not a top hit in the CRISPR screening. What is the rationale of selecting TCAF1 in this study? It would better to briefly describe the rationale behind the selection.
2. Line 118, "Table S1" should be Table S2.
3. Fig. 1D, representative fiber images should be shown in main figures.

4. Fig. 2C, statistical analysis should be performed to determine if the difference is significant.
5. Fig. 2F, Western blot should be performed to show the activation of AMPK signaling.
6. Fig. 2, the influence of TCAF1-depletion on Exo1 should be directly tested and shown.

Response to Reviewers (NCOMMS-23-44481)

Reviewer #1

In this manuscript, Kong et al identifies TCAF1 as a key factor promoting TRPV2-mediated Ca²⁺ release under conditions that activate cGAS/STING, highlighting the mechanism by which cytosolic DNA activates TRPV2. TCAF1 achieves this by facilitating the dissociation of STING from TRPV2, relieving TRPV2 repression. The authors further demonstrated that the role of TCAF1 is crucial for fork protection, chromosomal stability, and cell survival during replication stress. The manuscript is clearly written, the figures are presented in a logic way. Understanding how the TRPV2-mediated Ca²⁺ release pathway is activated by cytosolic DNA through the involvement of TCAF1 provides valuable insights into the molecular basis of genome maintenance and cancer prevention. This study thereby has the potential to impact our understanding of cancer biology and may contribute to the development of new strategies for cancer prevention and treatment. However, to further strengthen the molecular mechanisms presented in this study, additional evidence and explorations are warranted. For instance, it is crucial to investigate whether TCAF1 exhibits sensitivities to other types of DNA damage reagents, beyond HU. Moreover, exploring whether phenotypes observed in TCAF1 KD cells can be rescued by other replication or repair enzymes, besides EXO1, would provide a more comprehensive understanding. The most importantly, it is unclear what function of TCAF1 contributes to TRPV2 dissociation from STING. Mutants of TCAF1 which could not release TRPV2 should be identified for understanding the molecular mechanisms. Addressing these additional mechanistic insights and some methodological considerations will further strengthen its impact and contribution to the field.

We greatly appreciate the reviewer's enthusiasm for our manuscript and the potential impact of our study. We have addressed the reviewer's questions and suggestions in subsequent points.

Figure 1: Figure 1B raises questions regarding the sensitivity of TCAF1 to other forms of stress and DNA damage. It is essential to ascertain whether TCAF1 KD specifically sensitizes cells to replication stress or if it exhibits sensitivity to other DNA damage reagents.

In response to the reviewer's comments, we performed additional experiments to examine the sensitivity of TCAF1-knockdown (TCAF1-KD) HeLa cells to other genotoxic agents, including the radiomimetic drug bleomycin and the DNA topoisomerase I inhibitor camptothecin. We observed little or no sensitivity of TCAF1-KD cells to bleomycin or camptothecin (see **new Figure S1B**), suggesting that TCAF1 plays a unique role in the replication stress response.

Figure 2 shows phenotypes observed in TCAF1 KD cells can be rescued in EXO1 KD cells. It would be valuable to explore whether other fork protection enzymes can also rescue some of the phenotypes of TCAF1 KD cells.

As suggested by the reviewer, we conducted additional experiments to determine whether the other two major fork resection nucleases Mre11 and DNA2 are involved in the fork resection phenotype of TCAF1-KD cells. Our results shown in new Figure S2A in the revised manuscript indicate that the fork resection phenotype of TCAF1-KD cells was also rescued by Mre11 inhibition (by Mirin). In contrast, siRNA-mediated DNA2 depletion did not produce a similar rescue effect (see **new Figure S2B**). These results are consistent with the notion that Mre11 and Exo1 act in the same fork resection pathway, while DNA2's role in fork resection is

regulated by a distinct mechanism.

Figure 3 demonstrates that TCAF1 promotes TRPV2-mediated Ca²⁺ release after cGAS activation in response to cytosolic DNA utilizing TREX1 KD or plasmid transfection. However, there is a need to establish a connection between this phenomenon and replication fork protection, as cytosolic DNA can be induced by various other factors. It is essential to confirm whether this is a replication-specific reaction or a general response triggered by accumulated DNA damage.

We thank the reviewer for this insightful comment. Our prior work demonstrated that replication stress leads to the accumulation of cytosolic DNA, subsequently activating TRPV2-mediated Ca²⁺ release from the ER to protect replication forks (Li, Kong, *et al.*, *Mol Cell* 2023. PMID: 36696898). As we reported earlier and in the current study, cytosolic DNA induced by other conditions such as TREX1 KD or DNA transfection also activates TRPV2-mediated Ca²⁺ release. Moreover, we found that direct activation of cGAS (by MnCl₂) or STING (by cGAMP) also activates TRPV2-mediated Ca²⁺ release (current study). Our results shown in this study indicate that TCAF1 is required for TRPV2-mediated Ca²⁺ release induced by cytosolic DNA or direct cGAS/STING activation. In the replication stress response, this function of TCAF1 facilitates replication fork protection, through the activation of the downstream CaMKK2-AMPK-Exo1 pathway.

The physiological function of TCAF1 in TRPV2-mediated Ca²⁺ release induced by other conditions that cause cytosolic DNA or cGAS/STING activation is unclear at this point. Although this subject is beyond the scope of this manuscript, we are actively exploring the function of the TCAF1/TRPV2/Ca²⁺ pathway in other cellular responses such as innate immune response, autophagy, and senescence, all of which can be induced by cytosolic DNA or direct cGAS/STING activation.

Figure 5. Separation TCAF1-TRPV2 association and TCAF1-STING interaction will be important to prove the function of TCAF1. Elucidating the interaction between TCAF1 and different partners through truncations and mutants, coupled with structural analysis, would be instrumental in understanding how TCAF1 interacts with these partners and contributes to its function.

In response to the reviewer's comments, we performed additional co-IP experiments to further define the interactions between TCAF1-TRPV2 and TCAF1-STING. Our results indicate that the TCAF1-TRPV2 interaction was not affected by STING depletion (**new Figure 5E**). However, TRPV2 depletion partially reduced the TCAF1-STING interaction, suggesting a role of TRPV2 in bridging the TCAF1-STING association (**new Figure 5F**). In further support of this idea, we found that STING(FTW-AAS), which is deficient in TRPV2-interaction, also exhibited reduced TCAF1 binding (**new Figure 5G**). These data, in combination with other results described in the manuscript, further support the idea that TCAF1 and TRPV2 function as partners in STING dissociation and Ca²⁺ release after the binding of cGAMP to STING.

We agree with the reviewer that additional mutational analysis and structural studies will provide further insights into the interactions between TCAF1, TRPV2, and STING, and their significance in TCAF1's function; however, we hope the reviewer would agree with us that such studies are extensive undertakings, necessitating a substantial amount of time and effort, and would thus be more appropriately pursued as part of a subsequent project.

Other issues:

The authors described CRISPR KO of TCAF1 and EXO1 and clone selection. In the manuscript, all cell lines utilized siRNA mediated KD but not KO. It is important to address certain methodological inconsistencies, such as the use of siRNA-mediated KD rather than KO for cell lines in the manuscript.

We have used multiple methods, including siRNAs, shRNAs, and sgRNA/Cas9, to functionally disrupt TCAF1 in cancer cell lines and non-transformed cell lines, and have obtained consistent results regarding the effects on the cytosolic DNA/TRPV2-dependent signaling pathway and replication fork protection. The majority of our experiments were conducted using TCAF1 shRNAs and siRNAs, as detailed in the manuscript. We used CRISPR/Cas9-mediated KO as well as a functional rescue via TCAF1 re-expression to further validate the fork resection phenotype, presented in original Figure 1E. To avoid potential confusion, we have moved this result to **new Figure S1D** as additional evidence.

The authors should provide evidence of the interaction between STING(FTW/AAS) and TRPV2 through immunoprecipitation experiments in Figure 5F.

We previously identified STING(FTW/AAS) as a mutant that is deficient in TRPV2 interaction (Li, Kong, *et al.*, *Mol Cell* 2023. PMID: 36696898). In response to the reviewer's comment, we have added a co-IP result for STING(FTW/AAS)-TRPV2 in **new Figure S4C** in the revised manuscript as a control.

Reviewer #2

In this manuscript, You and colleague argue that TCAF1 promotes TRPV2-mediated Ca²⁺ upon the activation of cGAS-STING pathway. The authors use KO/KD methods then assess the effect of TCAF1. Although it is commendable that the authors used an array of measurements the effect is rather marginal. Below I list major issues:

We appreciate the reviewer's positive comment regarding our use of multiple measurements to delineate the role of TCAF1 in the STING/TRPV2/Ca²⁺-dependent pathway. We have addressed the reviewer's comments in subsequent points, including those on the quantification of results in certain figures.

- Authors have also generated knockout TCAF1 HeLa cells but still decided to use silenced TCAF1 cells throughout the manuscript. Why not use TCAF1 KO HeLa cells instead?

We have used multiple methods, including siRNAs, shRNAs, and sgRNA/Cas9, to functionally disrupt TCAF1 in cancer cell lines and non-transformed cell lines, and have obtained consistent results regarding the effects on the cytosolic DNA/TRPV2-dependent signaling pathway and replication fork protection. The majority of our experiments were conducted using TCAF1 shRNAs and siRNAs, as detailed in the manuscript. We used CRISPR/cas9-mediated KO followed by a functional rescue via TCAF1 re-expression to further validate the fork resection phenotype, presented in **original Figure 1E**. To avoid potential confusion, we have moved this result to **Figure S1D** as additional evidence.

- Although the effect of TCAF KO/KD is statistically significant, I am fully convinced with the significance - the difference is minor (often $\leq 20\%$ and heavily eschewed by outliers) in all the experiments presented in this

manuscript.

We identified TCAF1 in this study as a key component of the cytosolic DNA/TRPV2-dependent signaling pathway for fork protection through an unbiased genome-wide CRISPR screen and extensive follow-up validations and mechanistic studies. To ensure rigor, we also used multiple orthogonal methods to analyze specific cellular phenotypes including fork resection, Ca^{2+} release, AMPK activation, and protein-protein interactions. The effects of TCAF1's functional disruption can be influenced by several factors: the extent of TCAF1 depletion, variability among KO clones, and the specific phenotypes to be analyzed. Please also see below our response to comments on specific results.

- In Fig.1C and other quantitative analyses presented, it seems that the mean is shifted because of the outlier and there seems to be no difference between before and after silencing of TCAF1.

We have performed an outlier analysis (using the ROUT method, $Q=1\%$) for **Figure 1C**. After outlier removal, the **revised Figure 1C** shows that TCAF1-KD cells exhibited a significantly higher level of fork resection, which was rescued by the ectopically expressed Flag-TCAF1.

- Why did authors decide to use short hairpin technique in some experiments and silencing RNA in other in this manuscript?

We used shRNAs for stable TCAF1 depletion in most of the experiments in the manuscript. We used siRNAs in the analysis of the cell cycle (**Figure S1E**), AMPK activation (**Figures 2F, S2G-H, and 3G**), cGAMP production (**Figure 4A**), and protein-protein interaction (**Figures 5A-5C, S4A**), partly because of a higher efficiency of TCAF1 depletion we sometimes observed for the siRNAs used. A key function of TCAF1 is to promote the dissociation between STING and TRPV2 in response to cytosolic DNA or cGAS/STING activation. In response to the reviewer's question, we performed a co-IP experiment in shTCAF1 cells after MnCl_2 treatment and obtained a similar result to that in siTCAF1 cells (i.e. TCAF1 depletion rescued STING-TRPV2 association in the presence of MnCl_2) (see **new Figure S5B**). Note that some of the experiments were performed intentionally by different authors at different times using different TCAF1 knockdown approaches for result crosschecking. All the results obtained from these different approaches align with each other and support our conclusions.

- Authors should also include the Western blots for AMPK phosphorylation in Fig. 2F like in Chk1 phosphorylation.

As suggested, we have performed Western blots for AMPK phosphorylation as well as Chk1 phosphorylation in HeLa cells as well as MCF10A cells. Consistent with our IF result, the Western blot result shows that TCAF1 is required for AMPK phosphorylation induced by HU treatment in HeLa and MCF10A cells (**new Figures 2F and S2H**). We have moved the **original Figure 2F** (IF result) to **Figure S2** (**new Figure S2G**).

- Fig.S3A, fluorescence signal is very weak in shCtrl-shTrex1 which was not decreased further after silencing Trex1. This explains the poor difference in quantification results.

Our quantified results in **Figure 3A** show a clear reduction in GCaMP_{6s} signal (ER Ca^{2+}) in shCtrl-shTREX1

cells. We have updated the figure with new representative images for both shCtrl-shCtrl and shCtrl-shTREX1 samples. All the original images for **Figure 3A**, including the images shown in the **original Figure S3A**, will be uploaded onto the public data repository Dryad (datadryad.org). Please note that the representative images in the **original Figure S3A-S3D** have now been moved to **Figure 3** in the revised manuscript, as suggested by the reviewer in the next specific point.

- Authors should move the fluorescence images of Fig.S3 to Fig.3 to make it clearer.

As suggested, we have moved the representative fluorescence images in the **original Figure S3** to **Figure 3** to be together with the quantified results.

- Again in Fig.3D, the elevation is so minor that it seems like it's been caused by the outlier in quantification.

The effects of plasmid DNA transfection on Ca^{2+} release can be affected by the transfection efficiency. In response to the reviewer's comment, we repeated the experiment with a higher transfection efficiency. The new result shows a much larger increase in GCaMP6s signal after DNA transfection in shCtrl- cells, which was abolished by TCAF1-KD.

- Fig.3F, S3F, background signal is high after MnCl_2 treatment which is making it difficult to believe that after silencing TCAF1, phenotype is only partially rescued. Authors should keep similar baseline as in other figures.

In response to the reviewer's comments, we performed the MnCl_2 treatment experiments again and examined the effects of TCAF1 depletion on iCa^{2+} elevation (GCaMP6s) and ER Ca^{2+} release (GCaMPer). The new results are shown in **Figure 3E** and **Figure 3F**. The baseline signals of GCaMP6s are now similar to those in other figures. However, TCAF1 depletion (or TRPV2 depletion) only partially abrogated iCa^{2+} elevation after MnCl_2 treatment (**Figures 3F and S3B**). As we described in the manuscript, we think MnCl_2 treatment induces iCa^{2+} elevation via both TRPV2/TCAF1-dependent and -independent mechanisms. This means that in addition to TRPV2, other ion channels may also be activated by MnCl_2 . However, the ER Ca^{2+} release triggered by MnCl_2 was TRPV2/TCAF1-dependent, as shown by the GCaMPer results in both the **original and new Figures 3E and S3A**. Importantly, this TCAF1/TRPV2-dependent ER release of Ca^{2+} is responsible for downstream AMPK activation, as shown in **Figures 3G and S3C**.

- Again, as is throughout this manuscript, in Fig.4B, in the first lane vs 4th lane, shift in mean seems to be caused by the outliers in the 1st lane and there seems to be no difference if you remove those outliers.

We have performed an outlier analysis for the quantified result in Figure 4B. The effect of cGAMP transfection on the GCaMPer signal was actually "enlarged" after the removal of potential outliers (see **new Figure 4B**).

- Elevation of iCa^{2+} after STING depletion is quite weak in Fig S4C

In response to the comment, we performed the experiment again with a high STING knockdown efficiency, as detected by Western blot (**new Figure 4D**). This STING depletion resulted in an obvious elevation of iCa^{2+} , which was not affected by TCAF1 knockdown (**new Figure 4D**), indicating that TCAF1 is no longer required for TRPV2-mediated Ca^{2+} release when STING is absent. With the newly added Western blot result for STING

knockdown, we have removed the **original Figure S4**.

- Authors should move fluorescence images from Fig.S4 to Fig.4

As suggested, we have moved the representative images of GCaMP6s and GCaMP6r from the **original Figure S4** to **Figure 4**.

- PLA signal images are missing from Fig.5E and PLA result after MnCl₂ treatment is not shown contrary to being mentioned in the text.

Thank you for pointing this out. We have added the representative PLA images to **original Figure 5E**, which is presented as **Figure 5H** and **5I** in the revised manuscript. We have also included the PLA result to show the effects of MnCl₂ treatment on TCAF1-STING association (see **new Figure S4D**) (note that **original Figure S4** has been moved).

- Actin loading control is missing from Fig.5F and one should not see any STING being pulled down in lane 1 without TCAF.

We have repeated the co-IP experiment in the **original Figure 5F** and included a Western blot for Actin as a loading control for the Input samples. In this new experiment, we increased the stringency of co-IP by performing three additional rounds of washing for the anti-Flag immunoprecipitants, with the aim to further reduce nonspecific background signals. The new co-IP result with statistical analysis is shown in the **new Figure 5G**.

We have also repeat the co-IP experiment in the **original Figure 5D** and added a Western blot for Actin as a loading control for the Input samples (see **new Figure 5D**).

- No mention of Fig.S5 anywhere in the text.

Thank you for pointing this out. In the revised manuscript, we have cited all the supplemental figures, including the results in the original Figure S5, in the main text.

Reviewer #3

In this manuscript, the authors identified TCAF1 as a new player in the cGAS-STING-TRPV2-Ca²⁺-CAMMK2-AMPK-EXO1 pathway previously discovered by their lab. They showed that TCAF1 is required for TRPV2-mediated Ca²⁺ release by promoting STING-TRPV2 dissociation upon STING activation, supporting the idea that TCAF1 is a new node in this signaling pathway. The experiments were nicely done, and the conclusions are essentially supported by the data shown. However, this reviewer has a few minor concerns.

We greatly appreciate the reviewer's positive comments on our study.

1. TCAF1 apparently is not a top hit in the CRISPR screening. What is the rationale of selecting TCAF1 in this study? It would better to briefly describe the rationale behind the selection.

As described in the manuscript, we chose to study TCAF1 as a potential TRPV2 regulator in the Ca²⁺-dependent fork protection pathway for two major reasons. Firstly, our genome-wide CRISPR/Cas9 screen identified TCAF1—a known regulator of the TRPM8 channel—as a high-ranked gene that may promote cell survival during replication stress by suppressing Exo1-mediated fork processing. Secondly, TCAF1 has been shown to interact with other TRP channels, including TRPV6 and TRPM2, in addition to TRPM8 (Gkika et al., J Cell Biol 2015. PMID: 25559186). These observations prompted us to investigate the potential role of TCAF1 as a TRPV2 regulator in the Ca²⁺-dependent fork protection pathway. We have revised the text on Page 5 to further clarify the rationale for studying TCAF1.

2. Line 118, “Table S1” should be Table S2.

Thank you for pointing this out. We have made the correction.

3. Fig. 1D, representative fiber images should be shown in main figures.

We have added representative fiber images to **Figure 1D** as requested.

4. Fig. 2C, statistical analysis should be performed to determine if the difference is significant.

We have added the statistical analysis results to the cell survival results in **Figure 2C**, as well as **Figure 1B**, in the revised manuscript.

5. Fig. 2F, Western blot should be performed to show the activation of AMPK signaling.

As suggested, we performed Western blots for AMPK phosphorylation in HeLa cells as well as MCF10A cells. Consistent with our IF result, the Western blot results show that TCAF1 is required for AMPK phosphorylation after HU treatment (**new Figures 2F and S2H**). We have moved the **original Figure 2F** (IF result) to **Figure S2** (now **new Figure S2G**).

6. Fig. 2, the influence of TCAF1-depletion on Exo1 should be directly tested and shown.

As shown in **Figure 2A**, TCAF1-depletion apparently did not affect the protein level of Exo1.

REVIEWERS' COMMENTS

Reviewer #1 (Remarks to the Author):

The revised manuscript answered my questions. I recommend publication of "TCAF1 Promotes TRPV2-mediated Ca²⁺ Release in Response to Cytosolic DNA to Protect Stressed Replication Forks" in Nature communications.

Reviewer #2 (Remarks to the Author):

The authors have addressed all my concerns. I appreciate that they rigorously re-analyzed their results.

Reviewer #3 (Remarks to the Author):

The authors have nicely addressed my concerns in revision, and I have no more comments.

Response to Reviewers (NCOMMS-23-44481A)

Reviewer #1:

The revised manuscript answered my questions. I recommend publication of "TCAF1 Promotes TRPV2-mediated Ca²⁺ Release in Response to Cytosolic DNA to Protect Stressed Replication Forks" in Nature communications.

Reviewer #2:

The authors have addressed all my concerns. I appreciate that they rigorously re-analyzed their results.

Reviewer #3:

The authors have nicely addressed my concerns in revision, and I have no more comments.

We want to thank all the reviewers for their invaluable feedback and their recommendation for publication of our revised manuscript.